# A single amino acid polymorphism in a conserved effector of the multihost blast fungus pathogen expands host-target binding spectrum

Adam R. Bentham[1], Yohann Petit-Houdenot[2,3], Joe Win[2], Izumi Chuma[4], Ryohei Terauchi[5,6], Mark J. Banfield[1]*, Sophien Kamoun[2]*, Thorsten Langner[2]*

1 Department of Biological Chemistry, John Innes Centre, Norwich Research Park, Norwich, United Kingdom, 2 The Sainsbury Laboratory, University of East Anglia, Norwich Research Park, Norwich, United Kingdom, 3 Université Paris-Saclay, INRAE, AgroParisTech, UMR BIOGER, Thiverval-Grignon, France, 4 Obihiro University of Agriculture and Veterinary Medicine, Obihiro, Japan, 5 Kyoto University, Kyoto, Japan, 6 Iwate Biotechnology Research Center, Kitakami, Japan

* Mark.Banfield@jic.ac.uk (MJB); Sophien.Kamoun@tsl.ac.uk (SK); Thorsten.Langner@tsl.ac.uk (TL)

**Data Availability Statement:** The authors confirm that all data underlying the findings are fully available without restriction. Sequence information

## Abstract

Accelerated gene evolution is a hallmark of pathogen adaptation and specialization following host-jumps. However, the molecular processes associated with adaptive evolution between host-specific lineages of a multihost plant pathogen remain poorly understood. In the blast fungus *Magnaporthe oryzae* (Syn. *Pyricularia oryzae*), host specialization on different grass hosts is generally associated with dynamic patterns of gain and loss of virulence effector genes that tend to define the distinct genetic lineages of this pathogen. Here, we unravelled the biochemical and structural basis of adaptive evolution of APikL2, an exceptionally conserved paralog of the well-studied rice-lineage specific effector AVR-Pik. Whereas AVR-Pik and other members of the six-gene AVR-Pik family show specific patterns of presence/absence polymorphisms between grass-specific lineages of *M. oryzae*, APikL2 stands out by being ubiquitously present in all blast fungus lineages from 13 different host species. Using biochemical, biophysical and structural biology methods, we show that a single aspartate to asparagine polymorphism expands the binding spectrum of APikL2 to host proteins of the heavy-metal associated (HMA) domain family. This mutation maps to one of the APikL2-HMA binding interfaces and contributes to an altered hydrogen-bonding network. By combining phylogenetic ancestral reconstruction with an analysis of the structural consequences of allelic diversification, we revealed a common mechanism of effector specialization in the AVR-Pik/APikL2 family that involves two major HMA-binding interfaces. Together, our findings provide a detailed molecular evolution and structural biology framework for diversification and adaptation of a fungal pathogen effector family following host-jumps.

of all isolates used in this study are included in S1 Table. Protein structures, and the data used to derive these, have been deposited at the Protein DataBank (PDB) with accession codes 7NLJ (APikL2A/sHMA25) and 7NMM (APikL2F/ sHMA94). The full yeast two-hybrid HMA-domain library is available through Addgene (https://www. addgene.org/Sophien_Kamoun, plasmid accession numbers 168306-168456) and detailed information is available in Langner et al., (2021, Zenodo; https://zenodo.org/record/5148559#. YQQabC1Q1Xs).

**Funding:** Y.P-H. was supported by a young scientist contract from INRA. S.K. is funded by the Gatsby Charitable Foundation (https://www.gatsby. org.uk/) and UK Research and Innovation Biotechnology and Biological Sciences Research Council (UKRI-BBSRC) grant BB/P012574. M.J.B. is funded by the John Innes Foundation (https:// www.johninnesfoundation.org.uk/). S.K. and M.J. B. receive funding from UKRI-BBSRC grants BBS/ E/J/000PR9797 and BBS/E/J/000PR9798, and the European Research Council (ERC) proposal 743165. The funders had no role in study design, data collection and analysis, decision to publish, or preparation of the manuscript.

**Competing interests:** The authors have declared that no competing interests exist.

## Author summary

Plant pathogens secrete effector proteins inside the host where they interact with host target proteins to alter cellular processes and enable infection and disease progression. These effector proteins evolve rapidly to adapt to changing host targets and to avoid recognition by the plant immune system, especially in multihost pathogens that frequently undergo host jumps or host-range expansions. The blast fungus *Magnaporthe oryzae* is one of the most destructive disease of cereals. Host specialization of *M. oryzae* lineages is generally associated with gains and losses of effector genes and selection of adaptive mutations. However, the molecular mechanisms of how effector proteins adapt to host targets is poorly understood. Here, we unravelled the biochemical and structural basis underlying adaptive evolution of the effector APikL2. APikL2 belongs to a six gene family of sequence related effectors including the well-studied effector AVR-Pik. APikL2 is exceptionally conserved across host-specific lineages but displays host-specific polymorphisms. We combine biochemical, structural and evolutionary biology methods to provide a molecular evolutionary framework how this effector family adapts to changing host targets. In APikL2, a single amino acid change expands binding to a host-target of the heavy-metal associated domain containing protein family. By reconstructing the evolutionary history of the effector family, we show that this amino acid polymorphism is a derived trait. We combine the knowledge gained from our evolutionary and structural biology analyses to unravel a common mechanism of adaptation in the APikL effector family that involves diversification of two major target-binding interfaces. Together, our findings expand our knowledge about how pathogen effectors adapt to changing host targets following host jumps.

## Introduction

A number of plant pathogens evolve by occasionally shifting or jumping from one host plant to another. Such host jump events are expected to have a dramatic impact on the evolution of pathogen virulence effectors—secreted proteins that can translocate inside host plant cells to modulate host processes, suppress immunity and facilitate colonization [1–3]. As effectors become exposed to a different cellular environment, their evolution is driven by two broad selection pressures imposed by the new host plant [4]. Effectors can evolve to adapt to the cellular targets of their new hosts [1,5] or to evade detection by plant immune receptors that can detect them by direct binding or indirectly through their perturbation of host processes [6–10]. These co-evolutionary conflicts have generally led to expansive and highly diverse effector protein repertoires in plant pathogens, notably in fungal and oomycete pathogens (filamentous pathogens). Effector genes are often located in specific genomic regions that allow for rapid evolution, such as gene gain and loss, emergence of adaptive mutations and horizontal gene transfer [11–14]. In biotrophic filamentous plant pathogens, i.e. pathogens that infect and proliferate in living host tissue, accelerated gene evolution is mostly documented at the level of coevolution with a single host species, but it has also been associated with host-range expansions and host jumps [1,14–21]. However, the molecular and biochemical processes associated with adaptive evolution between host-specific lineages of a multihost plant pathogen remain poorly understood.

The blast fungus *Magnaporthe oryzae* (Syn. *Pyricularia oryzae*) is a plant pathogen best known as the causal agent of rice blast disease. However, despite its Linnean name, *M. oryzae* has an extensive host range comprising more than 50 grass hosts, including major cereal crops

such as wheat and millet as well as their wild relatives (https://ars-grin.gov/fungaldatabases/). Although *M. oryzae* as a species has dozens of host species, individual isolates tend to preferentially associate with a single grass genus, leading to genetically divergent, host-specific lineages [14,22,23]. The current view is that *M. oryzae* is an assemblage of genetic lineages under incipient speciation following host shifts or host range expansions [23–25]. Extensive genomic resources, including chromosome quality reference genomes of host-specific isolates, have enabled comparative genomics studies and make *M. oryzae* an excellent experimental system to investigate the link between effector diversity and host-specificity [26–28]. To date, over 20 *M. oryzae* effector genes have been experimentally validated, primarily for their avirulence (AVR) activities (activation of immunity), and large-scale effector resources are available [29]. The majority of the studied effector genes show presence/absence (P/A) polymorphisms as a consequence of gene gain/loss associated with colonization of particular host genera or species [14,22,30,31]. In addition, an excess of nonsynonymous substitutions, consistent with positive selection, has been reported for the effector genes AVR-Pik, AVR-Pizt, Slp1, PWL3, AVR-Pita3 and BAS2 [2,14], particularly within the rice-infecting lineage. These signatures of positive selection are generally viewed as adaptive and attributed to arms race interplay with the host plant immune system following the gene for gene model. Nonetheless, phylogenomic analyses that map key biochemical transitions of effector proteins onto a molecular evolution framework remain limited, especially at the species-wide level.

Filamentous pathogen effectors generally expand through duplication and diversification from common ancestors, and therefore tend to share similar features such as conserved structural folds [32]. In *M. oryzae*, a suite of secreted proteins known as MAX (*Magnaporthe oryzae* avirulence and Tox-B-like) effectors groups sequence-unrelated proteins that share a characteristic six-stranded beta-sandwich fold [33]. MAX effectors have expanded in *M. oryzae* accounting for ~5–10% of the total putative effector repertoire and about half of the experimentally validated effectors. Although ~100 MAX fold proteins have been identified by Hidden Markov Model (HMM) predictions [29,33], AVR-Pik, AVR1-CO39, AVR-Pib, AVRPiz-t and AVR-Pia remain the only *M. oryzae* MAX effectors that have been structurally and functionally characterised [8,10,14,34–38]. Three of these MAX effectors (AVR-Pik, AVR1-CO39 and AVR-Pia) bind host proteins carrying a heavy metal-associated (HMA) domain (see below) [10,34,36]. However, it remains unclear whether the MAX fold is predictive of the precise effector activities or whether it serves as a basic structural scaffold that confers protein structural integrity while providing a framework for sequence and functional diversification [14,33].

Although *M. oryzae* effectors have evolved to modulate host processes to benefit the pathogen, they have been initially discovered through their AVR activities or the capacity to activate immune receptors (encoded by resistance genes (R-genes) commonly called Pi genes) in resistant host genotypes. For example, the AVR-Pik and AVR1-CO39/AVR-Pia effectors activate the rice Pik and Pia immune receptor proteins, respectively, granting resistance to *M. oryzae* isolates carrying the cognate effector [39–42]. Both the Pik and Pia loci consist of paired intracellular sensor/helper receptors of the NLR (nucleotide-binding and leucine-rich repeat) class. Remarkably, both sensor NLRs (Pik-1 and Pia2 syn. RGA5) carry an extraneous integrated domain with similarity to the heavy metal-associated (HMA) proteins that are targeted by the effectors [10,34,36]. These integrated HMA domains of Pik-1 and Pia2 directly bind AVR-Pik and AVR-Pia/AVR1-CO39, respectively. Extensive structure-function studies of the integrated HMA domains in complex with the effectors demonstrated the importance of the binding affinity to the HMA domain for the activation of the NLR proteins [7,10,34,36,43,44]. Interestingly, comparative analyses of the crystal structures of AVR-Pik, AVR-Pia and AVR1-CO39 in complex with HMA domains revealed common and contrasting binding interfaces. Whereas

AVR1-CO39 and AVR-Pia bind to the Pia2-HMA at a single interface formed by hydrophobic interactions between residues of the β-2 strands of both the effector and the HMA [35], AVR-Pik binds to the Pik-1 HMA domain mainly by electrostatic interactions mediated by three distinct interfaces [7,10,43]. The AVR-Pik/Pik-1 HMA interfaces are comprised of main chain interactions between the residues of the C-terminal β-4 strand of the HMA and β-3 of the effector (interface 1), a salt bridge between the β-2 loop of the HMA and β-2 strand of the effector (interface 2), and interactions between the residues of the N-terminal loop of the effector and the β-2 and β-3 strands of the HMA (interface 3).

The remarkable finding that a single type of NLR-integrated domain mediates perception of three sequence diverse blast fungus effectors indicates that this pathogen has evolved multiple effectors to target HMA-containing proteins. Indeed, two recent studies validated the view that, before being baited by NLR proteins, AVR-Pik and other HMA binding effectors have convergently evolved to target host proteins of the heavy metal-associated plant proteins (HPPs) and heavy metal-associated isoprenylated plant proteins (HIPPs), collectively referred to as small HMA (sHMA) proteins [45–47]. sHMA proteins are metallochaperone-like proteins that belong to an expanded family of up to 100 members in a single grass species that contribute to metal homeostasis and detoxification during abiotic stress [45,48]. The exact role of sHMA proteins during plant-pathogen interactions is still unknown and is complicated by their genetic redundancy and complex interplay with multiple pathogen effectors. However, recurrent integration of HMA-domains into the canonical architecture of rice NLR proteins indicates that sHMA proteins are the operative targets of a number of *M. oryzae* MAX effector proteins. The relevance of sHMA proteins during pathogen infection is further supported by the finding that the rice sHMA proteins Pi21 and OsHIPP20 are susceptibility factors that support infection by *M. oryzae* [47,49].

The molecular processes associated with adaptive evolution in host-specific lineages of a multi-host plant pathogen are poorly understood. Although there is a wealth of knowledge about genome and effector gene evolution in the context of gene-for-gene co-evolution with plant immune receptors, there are only a few documented examples of pathogen effector adaptation to changing host target environments, particularly following host jumps and host shifts. Here, we investigated the biochemical and structural basis of adaptive evolution of APikL2 (syn. Pex75 as per [8,42]), an exceptionally conserved paralog of the well-studied effector AVR-Pik that is specific to the rice-lineage of *M. oryzae*. APikL2 stands out compared to the five other members of the AVR-Pik family by having limited presence/absence polymorphisms and being present in all *M. oryzae* lineages from 13 different host species. We combined biochemical, biophysical and structural biology methods to show that a single amino acid polymorphism expands the binding spectrum of APikL2 to a host sHMA protein. This mutation maps to an APikL2-HMA binding interface and contributes to an altered hydrogen-bonding network that may facilitate binding. Further, phylogenetic ancestral reconstruction and analysis of the structural consequences of allelic diversification of the AVR-Pik/APikL2 family revealed that two major HMA-binding interfaces have been targeted by positive selection. Altogether, our findings provide a detailed molecular evolution and structural biology framework for diversification and adaptation of a fungal pathogen effector during adaptation to a new host environment.

## Results

### A large family of effectors with sequence similarity to AVR-Pik occurs in multiple host-specific lineages of *Magnaporthe oryzae*

To identify AVR-Pik like (APikL) effector candidates across multiple host-specific lineages, we took advantage of 107 publicly available genome assemblies of *M. oryzae*, representative of ten

distinct genetic lineages including isolates from 13 host species (S1 Table). We also included six *Magnaporthe grisea* crabgrass (*Digitaria sanguinalis*) isolates and one fountain grass (*Pennisetum americanum*) isolate as outgroups in the analysis. The final set of *M. oryzae* genomes contained 24 *Oryza sativa*, six *Setaria spp.*, nine *Eleusine spp.*, 17 *Lolium spp.*, 37 *Triticum spp.*, one *Eragrostis curvala*, two *Brachiaria spp.*, two *Panicum repens*, and two *Stenotaphrum secundatum* infecting isolates from different geographical regions. To test whether these assemblies are suitable for presence/absence analyses, we performed a BUSCO analysis that confirmed completeness of all assemblies at >90% except the *M. grisae* isolates VO107 and Dig41 (BUSCO completeness of 84.4% and 88.7%, respectively).

We then identified APikL family members using an iterative TBLASTN approach. First, we used AVR-PikD (D variant of AVR-Pik; EnsemblFungi accession number in 70–15 assembly MG08: MGG_15972) as a query to identify sequence related proteins. This led to the identification of four effector candidates that we named APikL1, APikL2 (Syn. Pex75; accession number in assembly MQUD00000000.1 of isolate B71: BSY92_00423 according to [28]), APikL3, and APikL4. We then repeated the search using all APikL family members leading to the identification of APikL5, which is more divergent and carries a 19 amino acid C-terminal extension. In total, we identified 179 genes belonging to a family of six APikL effectors (including AVR-Pik; S1 File). Ten of these genes appeared to be pseudogenized in isolates belonging to the *Oryza*, *Setaria* and *Brachiaria* lineage 2. We defined genes as pseudogenes when one of the following conditions were met: i) mutated or missing start codon, ii) truncated sequence in the assembly or iii) premature stop codons present. The identified pseudogenes included two AVR-Pik genes that were C-terminally truncated in *Oryza* infecting isolates, one APikL5 gene that is N-terminally truncated in the *Brachiaria* infecting isolate Bd8401, four APikL4 genes in *Setaria* infecting isolates in which the start codon was mutated, and three APikL2 genes with premature stop codons in rice infecting isolates from the US. The exact same mutations were shared between all APikL4 or APikL2 pseudogenes, respectively. Commonly, isolates have one or two genes (or pseudogenes) of the family except isolate Bm88324 (*Brachiaria* lineage 1) and four *M. grisae* isolates isolated from *D. sanguinalis* that have no detectable APikL homologs.

Next, we mapped all APikL family members onto the genetic lineages of *M. oryzae*. We first generated maximum likelihood trees of 1920 conserved single copy orthologs of all isolates and applied a multispecies coalescence method using ASTRAL [50–52] to infer a coalescence species tree (Fig 1). The topology of the species tree was consistent with previously published analyses [23] and reflected the host-driven genetic specialization of *M. oryzae* lineages.

We then annotated presence/absence of each APikL family effector across the tree. Notably, most APikL family members showed lineage-specific presence/absence polymorphisms, similar to other host-specific AVR-effector proteins, with the following specificities: AVR-Pik was found in the *Oryza*-infecting lineage (and the *Panicum repens* isolates BTTrp-5 and -6), APikL4 in *Setaria*- and *Eleusine*-infecting lineages, and APikL5 in the *Lolium*-, *Triticum*- and *Eleusine*-infecting *M. oryzae* lineages (Fig 1). APikL1 and APikL3 were specific to the *Digitaria*- and *Pennisetum*-infecting species *M. grisea* and *M. pennisetigena*, respectively. Unlike the other APikL effectors, APikL2 was exceptionally conserved across all lineages irrespective of the host plant, with the exception of the *M. grisea* and *M. pennisetigena* isolates that represent the outgroup in our analysis.

## APikL2 is exceptionally conserved across all *M. oryzae* lineages but displays lineage-specific polymorphisms

In addition to presence/absence polymorphisms, allelic diversification is often associated with pathogen effectors. To determine the extent of host-lineage specific amino acid

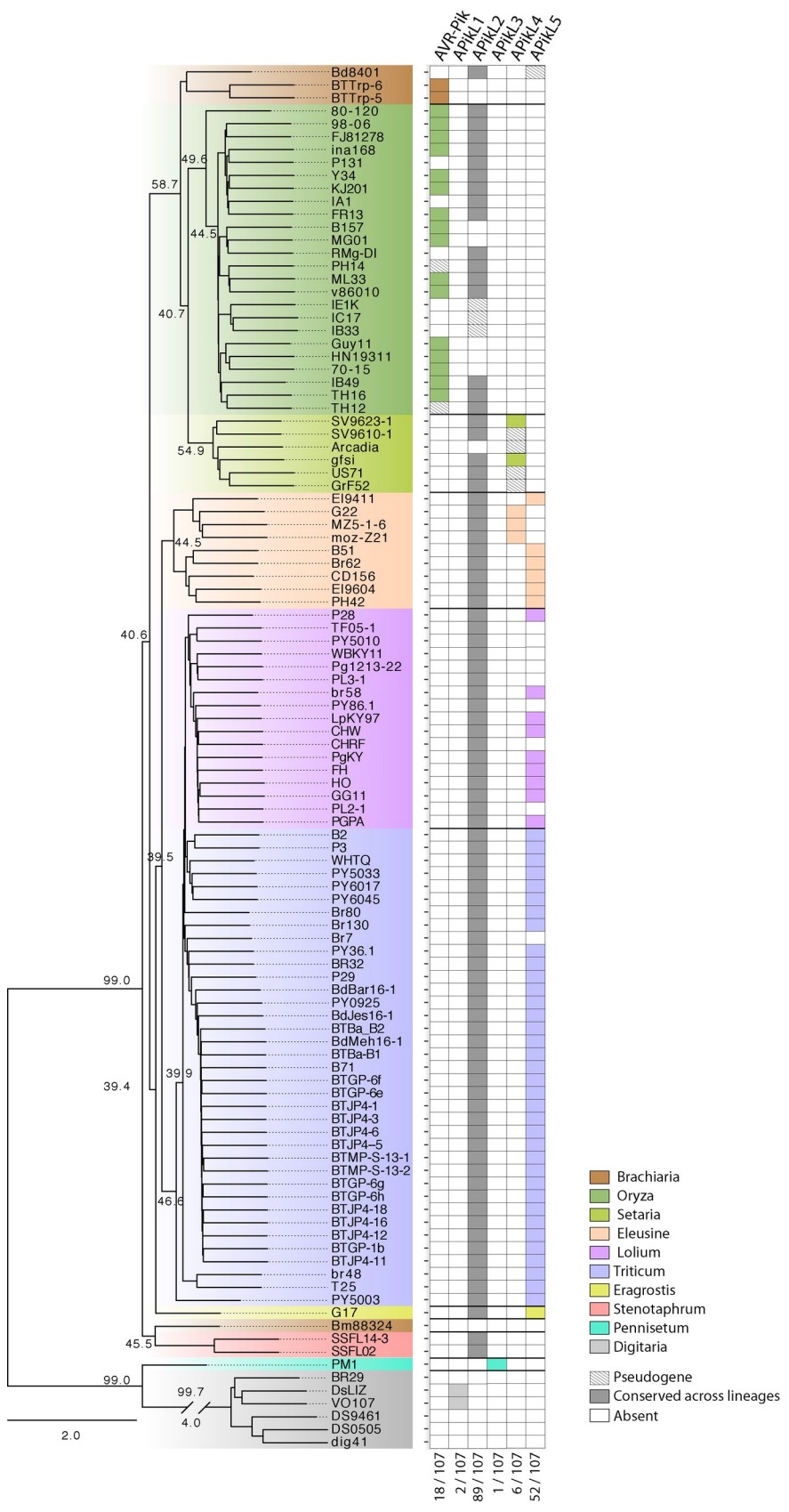

**Fig 1. APikL2 is conserved across different host-specific lineages of *M. oryzae*.** Presence/absence analysis of APikL family members shows conservation of APikL2 across all host-specific lineages of *M. oryzae*. All other family members show presence/absence polymorphisms that correlate with the species phylogeny and host-specificity. Left: ASTRAL multispecies coalescence tree derived from 1920 maximum-likelihood trees of all conserved single copy orthologs. Numbers at selected main branches represent the q1 local quartet tree support of individual genealogies. Right: Presence of APikL family members in each isolate. Colors indicate the host-specific lineages. The numbers at the bottom show the number of isolates that contain a certain effector.

polymorphisms in APikL effectors, we analysed allelic diversification within the APikL family. We used the full-length amino acid sequences of the 169 intact APikL family members to generate a maximum likelihood tree of the APikL family and map out allelic variants (Fig 2). We found one variant for APikL1 and APikL3, two variants for APikL4, four variants for APikL5, and six variants for APikL2 in addition to five known variants of AVR-Pik. The overall diversification of the whole APikL family resembled that of AVR-Pik which is known to be crucial for determining host compatibility driven by the co-evolution with the resistance locus Pik. Notably, we found that, although APikL2 is exceptionally conserved across all host-specific lineages of *M. oryzae*, the pattern of allelic diversification largely associated with particular lineages (Fig 2), similar to the pattern of presence/absence polymorphisms noted earlier for the rest of the APikL family.

## The APikL2 gene locates to a conserved region on chromosome 3 in diverse genetic lineages of *M. oryzae*

We explored whether the genetic conservation of APikL2 is reflected in its genomic environment. To gain insights into the genomic location and dynamics surrounding the loci of APikL family members, we took advantage of several near chromosome quality assemblies, including eight isolates representative of five genetic *M. oryzae* lineages that contain the APikL members AVR-Pik, APikL2, APikL4, and APikL5. We first generated whole genome alignments of each assembly using isolate 70–15 (EnsemblFungi assembly MG08 [26]) as reference. We then identified homologous chromosomes or contigs in all assemblies by pairwise comparison to 70–15 and mapped the genomic location of APikL genes to each alignment (S1 Fig). All analysed isolates contain the APikL2 gene in a region approximately 1.5 MB from the start of chromosome 3 (S1 Fig), except the reference genome of 70–15 or the closely related isolate Guy11 that have lost APikL2.

The region around the APikL2 locus displays intra- and inter-chromosomal inversions as well as deletions that likely resulted in loss of APikL2 in 70–15 and Guy11 (S1 Fig). However, these rearrangements do not affect the overall location of the APikL2 locus on chromosome 3. Conversely, the genomic location of other APikL members is more variable. AVR-Pik localizes to the end of chromosome 2 in 70–15, to a contig with local similarities to chromosome 1 in FJ81278, to both ends of the mini-chromosome of FR13 [53], and to the end of chromosome 7 in Guy11 that likely resulted from an inter-chromosomal translocation between chromosome 2 and chromosome 7. Intriguingly, APikL4 and APikL5 also locate to the end of chromosome 7 in the isolates MZ5-1-6 and CD156 or B71, respectively, whereas APikL4 localizes to chromosome 2 in the *Setaria* infecting isolate US71. This analysis revealed that the genomic location of APikL2 across the examined isolates is conserved, whereas other APikL genes locate to variable regions in the genome.

To gain a more detailed view of the conservation and synteny around APikL genomic loci, we extracted the homologous contigs containing APikL2 or other APikL family members and performed pairwise alignments (Fig 3). This analysis revealed that frequent inversions are characteristic of the APikL2 locus, whereas the majority of chromosome 3 is highly syntenic

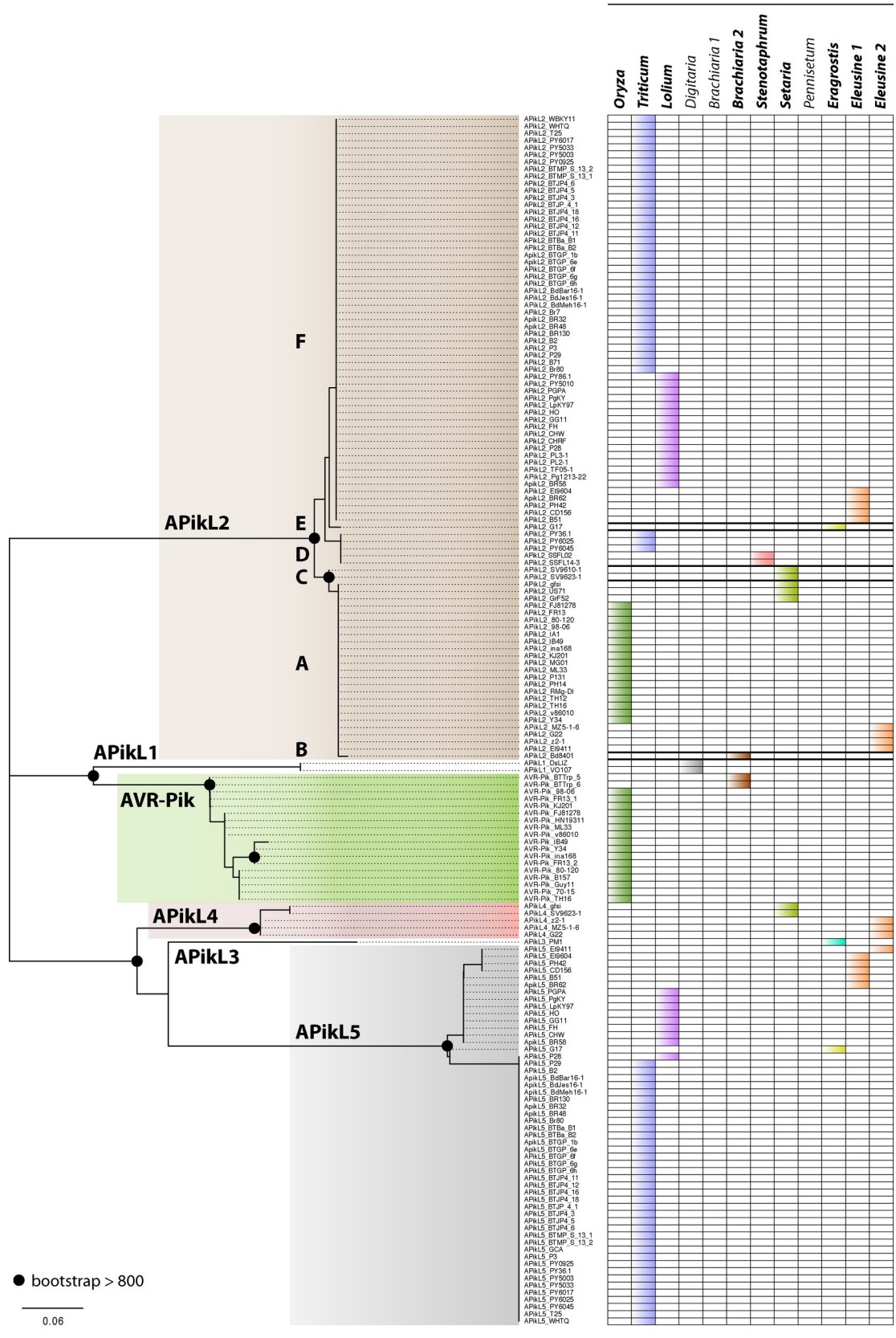

**Fig 2. APikL2 sequence diversification is associated with host-specific lineages.** Left: APikL family maximum likelihood tree based on amino acid sequences of all APikL proteins from 107 *M. oryzae* genomes. Colors indicate APikL family members where allelic diversification was observed in our dataset. Black circles indicate a bootstrap support >800 of the major nodes of allelic variants. Right: Presence of APikL family members in various host-specific lineages. Colors indicate host lineages as shown in Fig 1. Host-lineages in bold indicate presence of APikL2.

(Fig 3A). A similar pattern was present at the end of chromosome 7 where insertion and deletion events and an inversion surround the genomic region of APikL4, APikL5, and AVR-Pik (Fig 3C). We then extracted a 100 kb region surrounding the APikL loci and analysed the conservation and colinearity of all genes within the regions. We mapped the B71 gene models (according to [28]) to the extracted regions and visualized gene conservation. This analysis showed that APikL2 locates to a conserved region that contains 32 predicted genes from the B71 assembly. The majority of these genes are conserved and colinear between all analysed isolates and likely represent a secondary metabolite cluster (S2 Table). In isolates US71 (*Setaria* infecting lineage) and MZ5-1-6 (*Eleusine* infecting lineage), a ~30 kb insertion is present between APikL2 and the neighbouring gene BSY92_00422 (Fig 3B), but this insertion didn't disrupt the gene co-linearity. Conversely, APikL5 resides within a gene poor region in B71 containing 16 genes, mainly of unknown function. Eight and nine of these genes are conserved between B71 and CD156 or MZ5-1-6, respectively. Even though, AVR-Pik also locates to the end of chromosome 7 in Guy11, none of the B71 genes from the APikL4 or APikL5 loci is present in the 100 kB region surrounding the effector gene (Fig 3D). In summary, these analyses showed that APikL2 resides in a conserved region on chromosome 3 across diverse host-specific isolates, whereas the genomic environment surrounding other APikL family members is markedly more variable.

## APikL2 displays patterns of positive selection

Effector diversification is often driven by co-evolution of the pathogen with its host, which imposes strong selection pressure on the pathogen. To gain insights into the evolution and putative host-selective pressure acting upon the APikL family we generated multiple sequence alignments, reconstructed the APikL phylogeny and inferred the selective pressures acting on APikL family members. We applied branch models and pairwise comparison implemented in the package PAML v4.9 [54,55]. These detected signatures of positive selection in the branches leading to APikL2 ($\omega$ ($dN/dS$) = 1.27/1.47; free-ratio model/multiple-ratio model), APikL1 ($\omega$ = 1.75/1.43), and APikL5 ($\omega$ = 1.19/1.05; Fig 4A) consistent with high $dN/dS$ (ratio of non-synonymous and synonymous substitutions) in pairwise comparisons (Fig 4B).

Importantly, as previously observed for pathogen effector genes under extreme levels of selection [56–61], several terminal branches showed amino acid substitutions in the absence of synonymous changes ($dS$ = 0; Fig 4B and 4C). The underlying reason for these $dN/dS$ patterns is thought to be the extreme levels of selection imposed by the host and associated population bottlenecks (as discussed by [14]).

## Differential interaction of APikL2 variants with the HMA domain protein sHMA94

Given the APikL2 sequence similarity to AVR-Pik, we hypothesized that it binds plant HMA domains. We therefore tested interaction of APikL2A and AVR-PikD with a collection of 151 HMA domains from *Oryza*, *Setaria* and *Triticum* in a pairwise yeast two-hybrid screen. These included 42 and 96 HMA domains from *S. italica* and *O. sativa*, respectively, representative of the sHMA diversity in these species and 13 HMA domains from *T. aestivum* with high

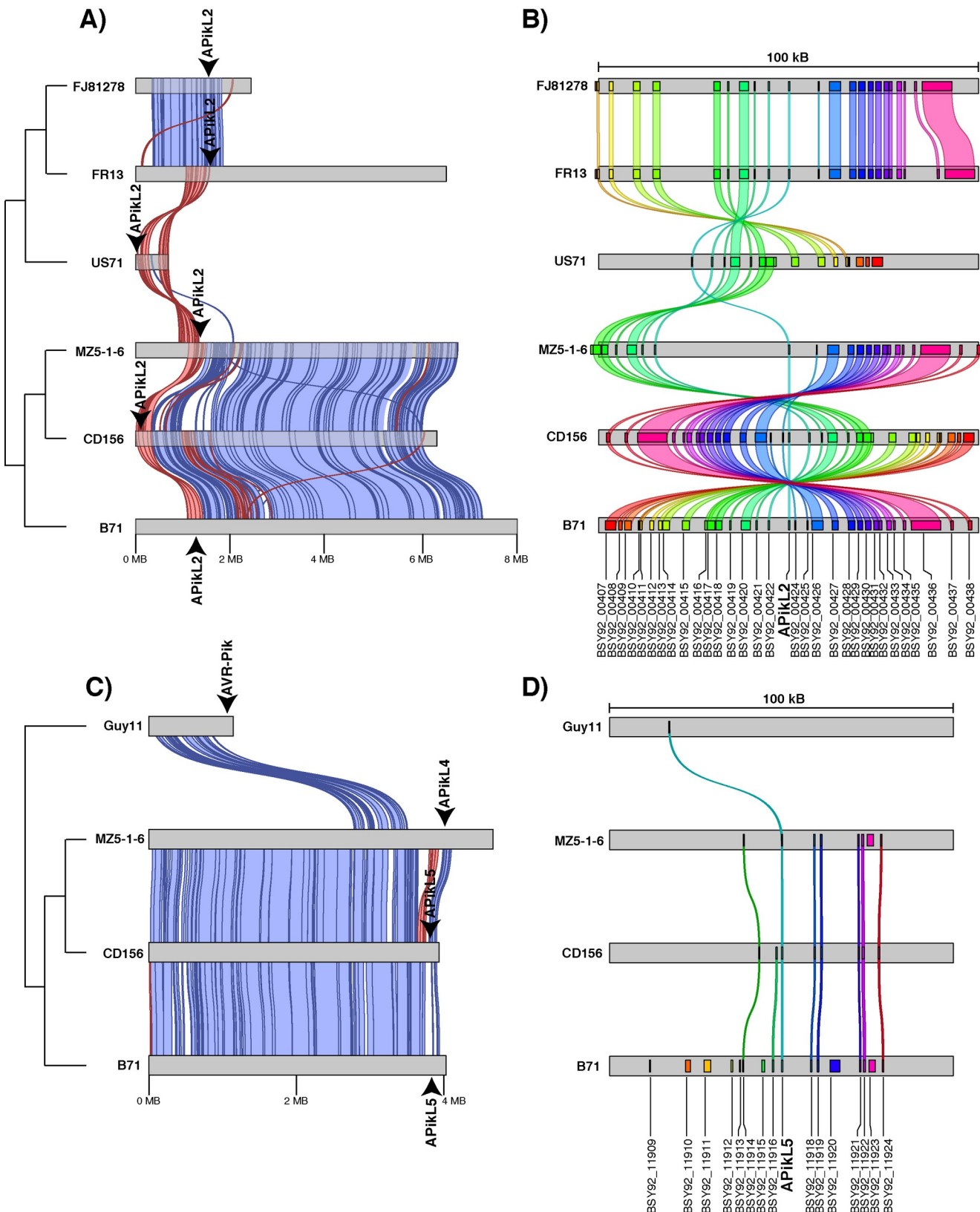

**Fig 3. The APikL2 locus is conserved across diverse genetic lineages of *M. oryzae*.** Synteny analysis of APikL loci in six isolates from five genetic *M. oryzae* lineages. **A)** Pairwise alignments of contigs containing the APikL2 gene. Forward and reverse alignments are shown in blue and red, respectively.

Arrowheads indicate the position of the APikL2 genes in the assemblies. **B)** Conservation of genes in a 100 kB region surrounding the APikL2 locus. Colored links indicate homologous genes. **C)** Pairwise alignments of contigs containing diverse APikL family members on chromosome 7. Forward and reverse alignments are shown in blue and red, respectively. Arrowheads indicate the position of APikL family genes in the assemblies. Genetic relationship of the isolates is shown schematically on the left. **D)** Conservation of genes in a 100 kB region surrounding the loci of APikL family members on chromosome 7. Colored links indicate homologous genes.

similarity to *O. sativa* clade A sHMAs [47]. This screen revealed 19 and 56 interactions for APikL2A and AVR-PikD, respectively (S3 Table). APikL2A interacted with 3, 5, and 11 HMA domains from *S. italica*, *T. aestivum*, and *O. sativa*, respectively. AVR-PikD interacted with 9, 7, and 40 HMA domains from *S. italica*, *T. aestivum*, and *O. sativa*, respectively. Overall, this screen revealed overlapping HMA-binding spectra of AVR-PikD and APikL2A.

Next, we asked whether host-specific variants of APikL2 differentially interact with HMA domains. APikL2 is invariant in *Oryza*-infecting isolates but diverged into six variants across host-specific lineages of *M. oryzae* (Fig 2). The most divergent variant to APikL2A is APikL2F, present in *Oryza*- and *Triticum/Lolium*-infecting lineages, respectively (Figs 2 and 4). We therefore tested differential interaction of APikL2A and APikL2F with the HMA domains from *S. italica* and *T. aestivum* in a pairwise yeast two-hybrid assay. This screen revealed that APikL2A and APikL2F differentially interact with one target HMA-domain from the host plant *S. italica*, Si944233294 (EnsemblPlants accession: SETIT_012363mg), designated from here on as sHMA94 (Fig 5). In yeast two-hybrid assays, sHMA94 showed strong interaction with APikL2F but no detectable interaction with APikL2A, which might reflect functional specialization between the two effector variants. Three additional HMA domains from *S. italica* (Fig 5) and five HMA domains from *T. aestivum* interacted with both APikL2 variants (S3 Table).

## Two polymorphic residues in APikL2 define interaction specificity with sHMA94

Four amino acids are polymorphic between the APikL2A and APikL2F proteins. To analyse the effect of these polymorphisms on the differential interaction to sHMA94, we tested 12 out of 16 possible mutant combinations of APikL2A and APikL2F in pairwise yeast two-hybrid assays against sHMA94 (Fig 6). We also included the HMA domain of Si514802025 (hereafter called sHMA25; EnsemblPlants accession: SETIT_011333mg), which interacted with both APikL2 variants in our previous screen, as a control to exclude that any observed effects are due to secondary effects, such as protein stability.

These assays revealed that two polymorphic sites at position 66 (Asp in PikL2A vs. Asn in APikL2F) and 111 (Leu in APikL2A vs. Pro in APikL2F) are crucial determinants for sHMA94 interaction specificity of APikL2 variants (Fig 6). Mutants that carry the APikL2F residues Asn-66 and Pro-111 (M4, M6 and M7 in Fig 6) interacted with sHMA94 independently of the other two polymorphisms. In addition, mutants that carry Asn-66 but retain the APiKL2A residue Pro-111 (M1-3 and M5 in Fig 6) displayed a weak interaction with sHMA94 further highlighting the importance of this residue.

## Asn-66 facilitates APikL2 binding to sHMA94

To validate the yeast two-hybrid assays and further investigate the affinity of APikL2A and APikL2F for sHMA94, we performed isothermal titration calorimetry (ITC) with recombinantly expressed proteins purified from *Escherichia coli* (Fig 6B). The change in heat ($\Delta$Q) for the interaction between APikL2A and sHMA94 was notably reduced compared to the interaction between APikL2F and sHMA94, suggesting a significant change in the thermodynamics

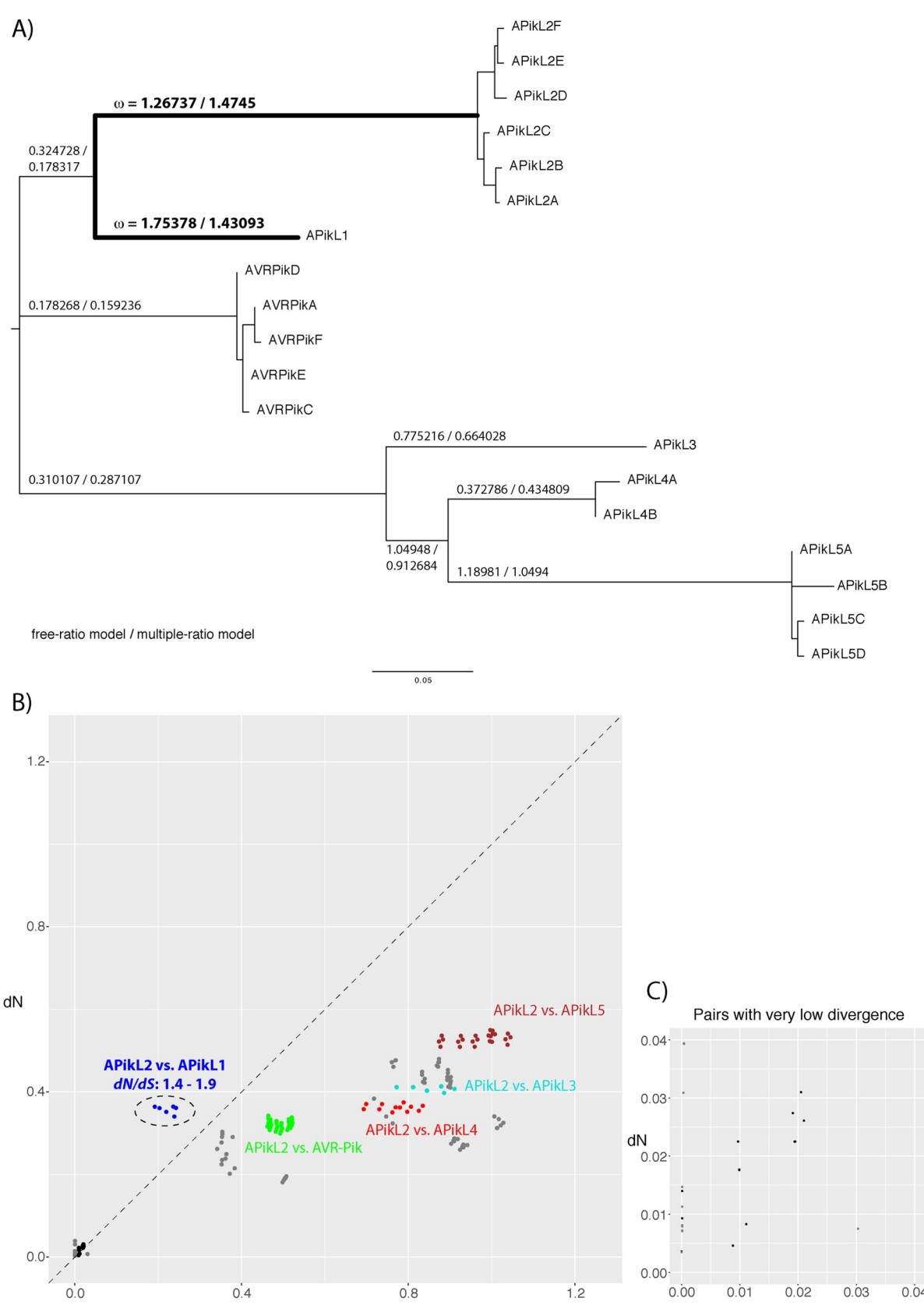

**Fig 4. APikL2 displays patterns of positive selection. A)** Maximum-likelihood tree of all APikL family members based on nucleotide sequences. Numbers on branches of the tree represent ω (*dN/dS* ratio of non-synonymous and synonymous substitutions) as calculated according to the free-ratio and multiple-ratio model implemented in PAML. Branches with high ω leading to APikL2 and APikL1 are highlighted in bold. **B)** Comparison of pairwise *dN/dS* ratios between APikL family members. Colors indicate pairwise comparisons between APikL2 and other APikL family members. *dN/dS* ratios between other APikL family members are shown in grey. Pairwise comparisons between APikL2 variants are shown in black. The dotted line indicates *dN/dS* = 1. **C)** Enlarged view of pairwise comparisons in B with low divergence.

of the interaction. Consequently, we could only measure binding affinity of APikL2F to sHMA94 with a calculated $K_d$ of 145.8 nM. For APikL2A we could not determine an accurate $K_d$, however, it is possible that it interacts weakly with sHMA94.

Next, we tested the influence of the key polymorphisms identified in the yeast-two hybrid mutant screen on in vitro sHMA94 binding. We first measured the effect of the Asp to Asn mutation at position 66 in APikL2, as this is the only polymorphism between APikL2A and APikL2F that is predicted to locate to the HMA binding interface (based on [7,10,46]). Introduction of Asn-66 into APikL2A was sufficient to facilitate sHMA94 binding, with a $K_d$ of 384.6 nM similar to APikL2F (Fig 6B). By contrast, the Asp-66 mutant reduced the affinity of APikL2F for sHMA94, resulting in no measurable binding of sHMA94, similar to APikL2A. Furthermore, the ΔQ of the interactions between the mutants and sHMA94 mirrored the changes in affinity, with the interactions between APikL2A Asn-66 mutant and sHMA94 exhibiting a larger ΔQ, akin to wild-type APikL2F, and APikL2F Asp-66 mutant showing a significantly reduced ΔQ, similar to wild-type APikL2A. While the yeast two-hybrid results also suggested an important role for Pro-111 (L111P in mutants M4, M6, M7) in sHMA94 interaction, we did not test this mutant in ITC as the single Asp-66 to Asn-66 mutation in APikL2A

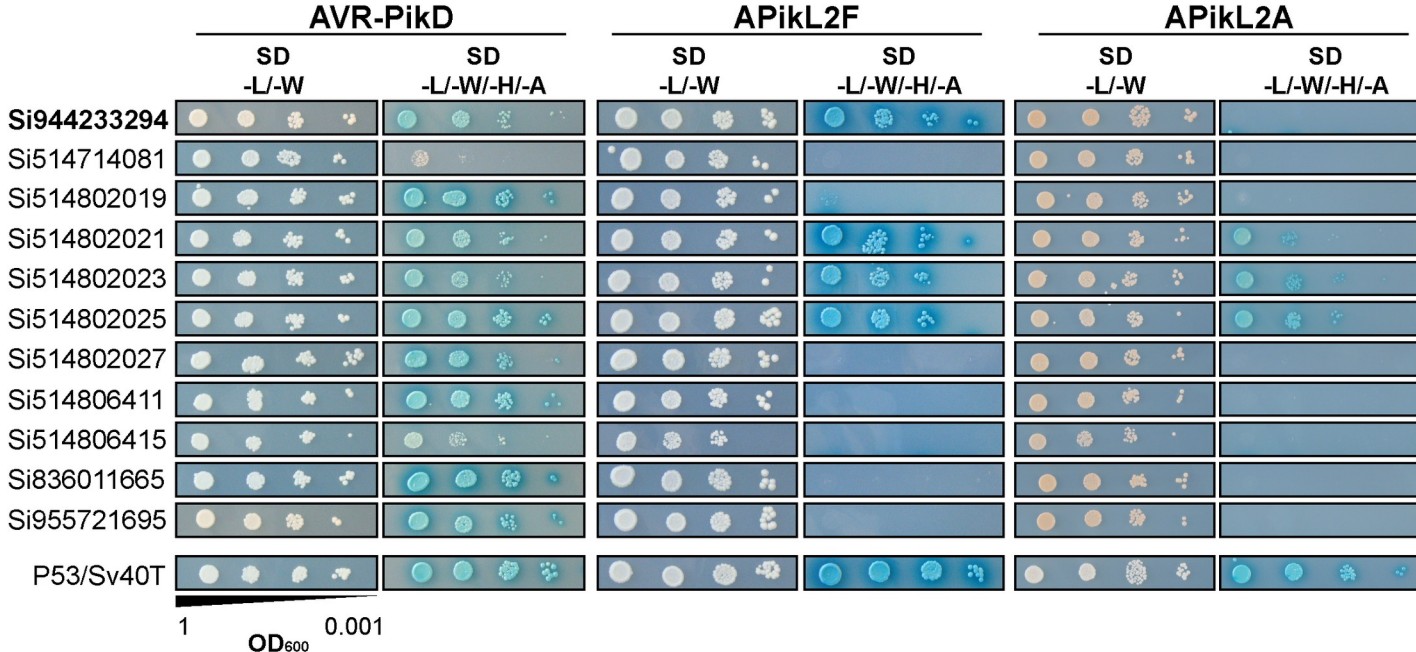

**Fig 5. APikL2 variants interact differentially with the HMA-domain of *Setaria italica* protein sHMA94.** Subset of the pairwise yeast two-hybrid screen to identify interaction partners of AVR-Pik and the APikL2 variants APikL2A and APikL2F from the *Oryza-* and *Triticum*-infecting lineages of *M. oryzae*. All positive interactions between *S. italica* HMA-domains with the APikL family members AVR-Pik and APikL2 are shown. Left: gene identifiers of HMA-domain containing proteins. SD -L/-W: Synthetic double dropout medium lacking the amino acids leucin and tryptophan; SD -L/-W/-H/-A: Synthetic quadruple dropout medium lacking the amino acids leucine, tryptophan, histidine, and adenine.

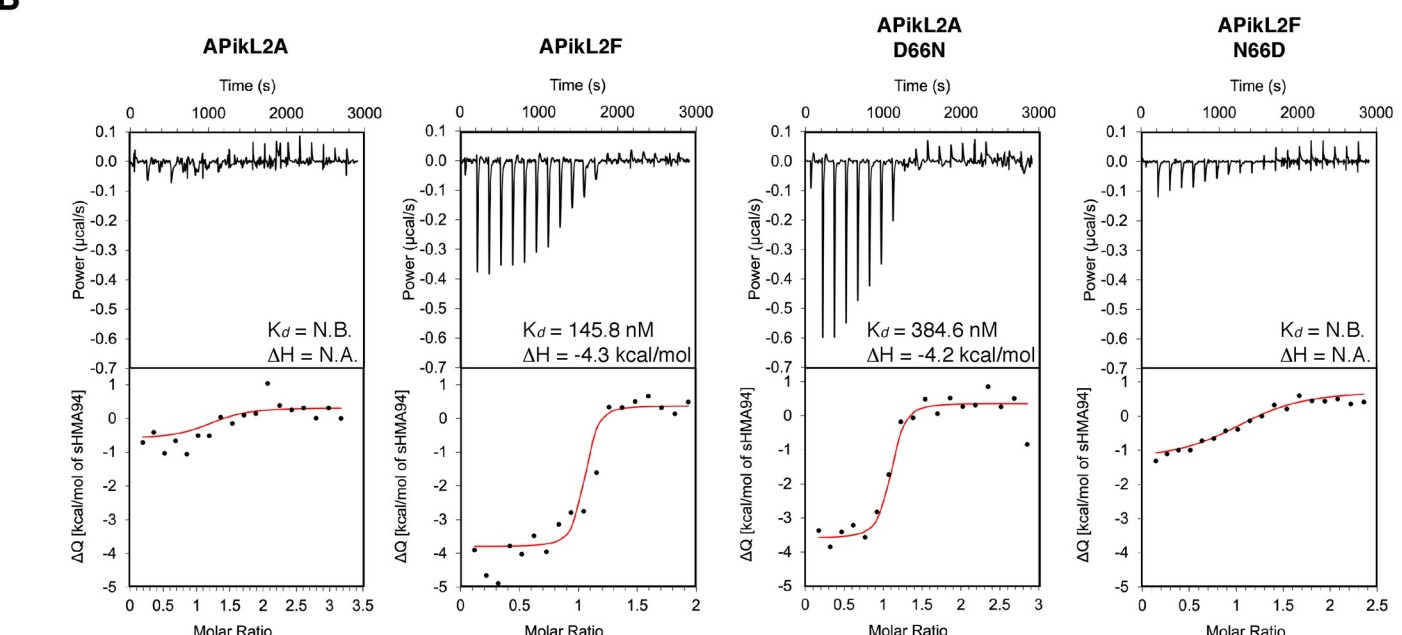

**Fig 6. Two polymorphic sites determine binding specificity of APikL2 variants to sHMA94. A)** Pairwise yeast two-hybrid assays between sHMA94 and APikL2A, APikL2F, and 12 APikL2 mutants derived from combinations of the four polymorphic residues. The two naturally occurring variants, APikL2A and APikL2F, were used as positive and negative controls. Mutations in each variant are shown schematically (left). sHMA25 was used as a positive control for both APikL2 variants. SD -L/-W:

Synthetic double dropout medium lacking the amino acids leucine and tryptophan; SD -L/-W/-H/-A: Synthetic quadruple dropout medium lacking the amino acids leucine, tryptophan, histidine, and adenine. **B)** Asn-66 is essential for binding of APikL2 to sHMA94 in vitro. In vitro binding affinities for sHMA94 and APikL2 proteins were assessed by ITC. The top panels show heat differences upon injection of the effector into the cell containing sHMA94. The lower panel shows integrated heats of injection (•) and the best fit (red line) to a single site binding model calculated using AFFINImeter ITC analysis software. APikL2F, but not APikL2A, binds sHMA94. The APikL2F Asn-66 to Asp-66 polymorphism disrupts sHMA94 binding. Conversely, the APikL2A Asp-66 to Asn-66 polymorphism facilitates binding of sHMA94, with affinity similar to APikL2F.

restored sHMA94 binding to similar levels as APikL2F. From these data, we conclude that Asn-66 in APikL2 facilitates binding to sHMA94 compared to Asp-66.

## Crystal structures reveal that Asn-66 stabilizes the binding interface between APikL2 and sHMA94

To better understand the molecular basis of interactions between APikL2 variants and sHMA proteins we sought to determine crystal structures of complexes. We crystallized complexes of APikL2F with sHMA94 and APikL2A with sHMA25 following co-expression in *E. coli* for comparative structural analysis. We were unable to crystallize a complex of APikL2A with sHMA94 for further study. X-ray diffraction data were collected at the Diamond Light Source (Oxford, UK) to 2.3 Å and 1.8 Å resolution, for APikL2F/sHMA94 and APikL2A/sHMA25, respectively. The structures were solved and refined as described in the Materials and Methods (S4 Table). In each case, the APikL2 variants and HMAs form 1:1 complexes, with both displaying the same global fold and interaction interfaces as previously described for the interaction of AVR-Pik with the Pik-1 integrated HMA domains [7,10] (Fig 7). One exception is the presence of an ordered N-terminal helix for APikL2, which is positioned away from the HMA-binding interface, but is disordered in AVR-Pik. The APikL2F/sHMA94 and APikL2A/sHMA25 complexes are very similar to each other (they superimpose with root-mean-square-deviation (RMSD) of 0.582 Å across the aligned Cα backbones). Further, both complexes are very similar to the structure of Pikp-1 HMA/AVR-PikD (PBD ID: 6G10) and could be superimposed on this structure with RMSD's of 0.9 Å and 1.04 Å, respectively.

As we were unable to crystallize the complex of APikL2A with sHMA94, we compared the APikL2F/sHMA94 and APikL2A/sHMA25 structures here (sHMA25 interacted with both APikL2 variants in our yeast two-hybrid assays). The APikL2F/sHMA94 and APikL2A/sHMA25 complexes display the three binding interfaces previously reported for AVR-Pik/HMA complexes [7,10]. They share similar interface areas of ~964 Å and ~1011 Å (calculated by QtPISA [62]) contributed by 57 and 56 residues, respectively. They also form a similar number of hydrogen bonds, salt bridges, and maintain a similar predicted binding energy (S5 Table). The largest differences between the complexes are within interface 3, where APikL2 forms several interactions with sHMA25 that cannot be formed with sHMA94 due to differences in the sequences of the HMAs in this region (S2 Fig). The interface 2 region of sHMA94 and sHMA25, which forms contacts with the Asp-66/Asn-66 residues of APikL2, is highly conserved in both complexes.

However, we observed that the presence of Asn-66 in APikL2F results in an altered local hydrogen bonding network around Gln-44 in sHMA94, when compared to Asp-66 in the APikL2A/sHMA25 complex (Fig 8). As a result, the side chain of Gln-44 of sHMA94 is rotated relative to Gln-43 of sHMA25, allowing Gln-44 to form additional intramolecular hydrogen bonds with Thr-38 in sHMA94 (Fig 7B and 7D). In this orientation, the NE2 atom of Gln-44 interacts with the OG1 atom and backbone carbonyl of Thr-38. This hydrogen bonding arrangement is not adopted in the APikL2A/sHMA25 complex, as the Asp-66 in APikL2A forms a hydrogen bond with the NE2 atom of Gln-43 in sHMA25 (Figs 7A, 7C and S3), rotating the NE2 group away from Thr-37, and preventing formation of hydrogen bonds between

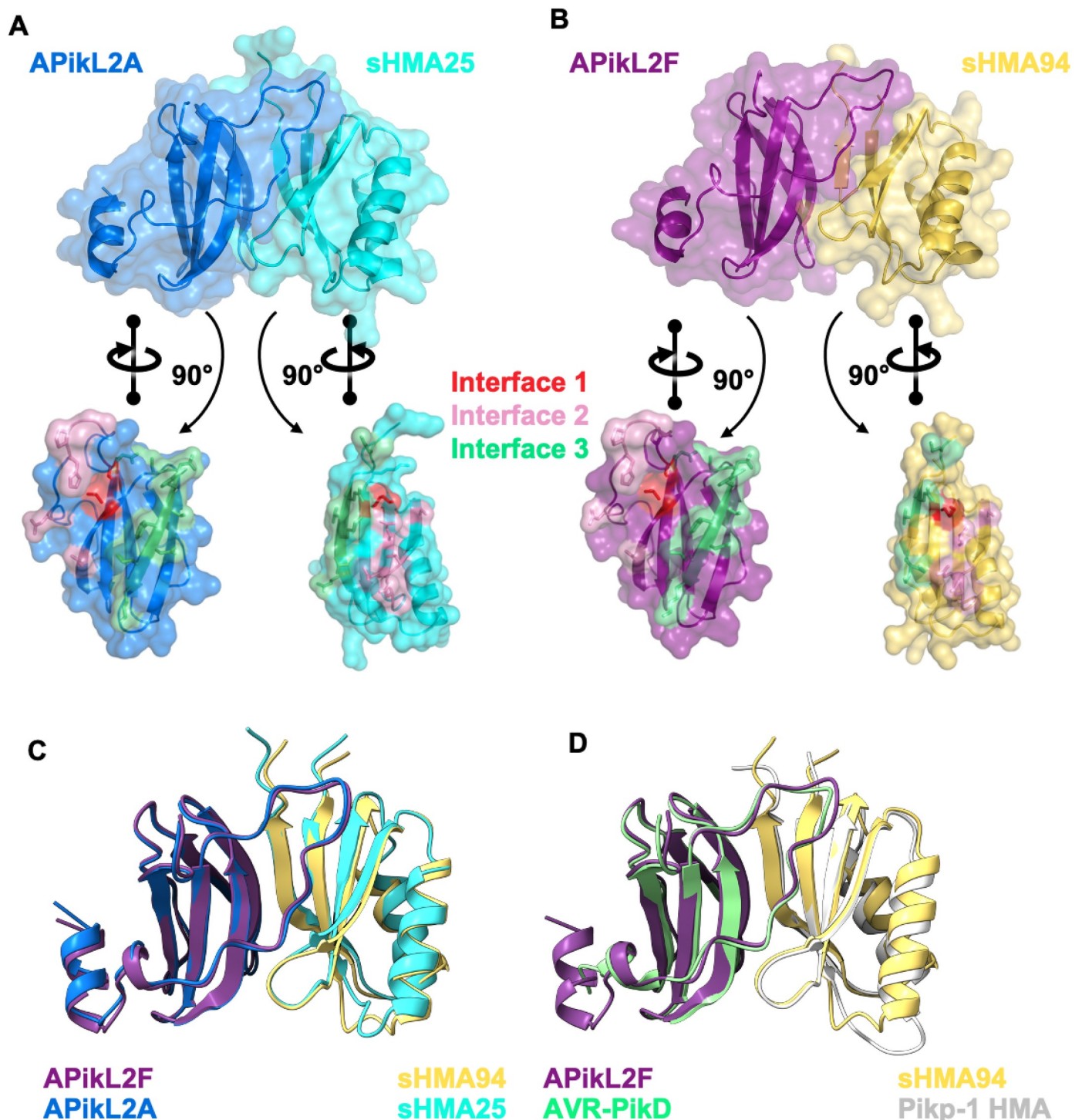

**Fig 7. Crystal structures of APikL2 in complex with HMA domains reveals a conserved binding interface. A)** Cartoon and surface representation of the APikL2A/sHMA25 complex, with APikL2A coloured blue and sHMA25 coloured cyan. **B)** Cartoon and surface representation of the APikL2F/sHMA94 complex, with APikL2F coloured purple and sHMA94 coloured yellow. The effector was separated from the complex using PyMol and rotated 90˚ relative to the complex to give an interior view of the interaction interface. The residues that form the interfaces one, two, and three (as defined by [7]), between the effector and HMA are coloured dark blue, pink, and magenta, respectively. **C)** Superimposition of the APikL2F/sHMA94 (purple and yellow) and APikL2A/sHMA25 (blue and cyan) complexes performed with PyMol. **D)** Structural alignment of APikL2F/sHMA94 (purple and yellow) with AVR-PikD/Pikp-1 HMA (green and grey; PDB ID: 6G10).

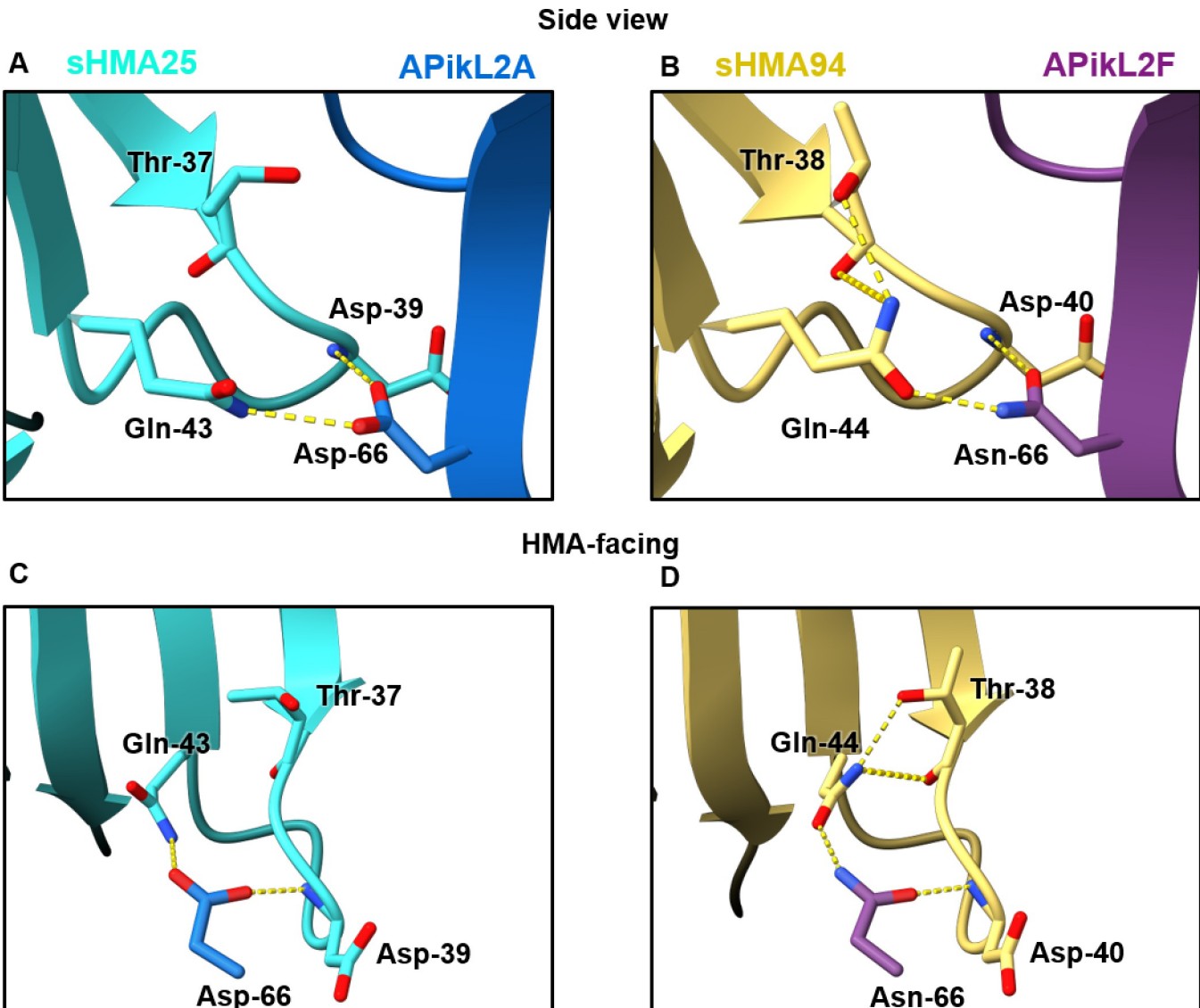

**Fig 8. The aspartate/asparagine polymorphism at position 66 in APikL2 results in an altered hydrogen bonding network at the sHMA interface. A, C)** The carboxylic side chain of Asp-66 from APikL2A forms two hydrogen bonds with sHMA25 through the NE2 atom of Gln-43 and the backbone nitrogen of Asp-39. **B, D)** The amide side chain of Asn-66 from APikL2F forms two hydrogen bonds with sHMA94 through the OE1 atom of Gln-44 and the backbone amide of Asp-40. The interaction with Gln-44 via the OE1 atom in the AVR-Pik2F/ sHMA structure flips Gln-44 respective to the Gln-43 residue of the APikL2A/sHMA25 structure. This reorientation of Gln-44 in sHMA94 allows the NE2 atom of Gln-44 to form intramolecular hydrogens bond with the OG1 atom and backbone carbonyl of Thr-38.

these two residues. The altered hydrogen bonding network around Gln-44 in the APikL2F/ sHMA94 complex, compared to APikL2A/sHMA25, appears to firmly lock Gln-44 in place compared to Gln-43 of sHMA25. This suggests that the interaction with Asn-66 of APikL2F results in reduced flexibility of the sHMA94 loop in interface 2. Our data indicates that this hydrogen bonding network stabilizes the interaction between APikL2F and sHMA94, providing an explanation for the binding specificity observed in the yeast two-hybrid and *in vitro* binding affinity assays. However, it is likely that the three interfaces differentially impact binding to divergent HMA-domains. Consistent with this, both APikL2 variants can bind to

sHMA25 where hydrogen bonds in interface 1 and 3 must have a stronger impact on complex formation compared to sHMA94.

### Asn-66 is a recently derived polymorphism in APikL2 effectors

Our results show that APikL2 and AVR-Pik share a common fold and HMA domain binding interfaces. This suggests that all APikL family members share a common fold and that specialization to host target HMAs is facilitated by adaptive mutations in common interfaces. To investigate this, we first mapped the evolutionary history of the polymorphisms that define the effector diversity in natural populations of *M. oryzae* and reconstructed the ancestral states of APikL family members (Fig 9A). This revealed that successive mutations have led to host-specific alleles for each of the APikL effectors. Consistent with tests for positive selection, most of the changes that contribute to allelic diversity in terminal branches are not associated with synonymous changes, similar to AVR-Pik and other AVR-effectors under strong positive selection [56–61]. The ancestral reconstructions further revealed the stepwise order of emergence of the crucial polymorphisms that determine binding specificity in both APikL2 and AVR-Pik (Fig 9A). The ancestral state at position 66 is Asp, which only mutated to Asn-66 in the branch leading to APikL2F, APikL2D and APikL2E (Fig 9A). In contrast, Pro-111 is ancestral to Leu-111, which only emerged in the APikL2A and APikL2B branch (Fig 9A). These results indicate that APikL2 binding to sHMA94-like HMA domains is a recently derived trait that evolved through the Asp to Asn polymorphism at position 66. We conclude that the most plausible evolutionary scenario is that Asn-66 enabled the APikL2 effector to expand its binding spectrum to sHMA94-like target proteins.

### Diversifying residues are centred around the two HMA-binding interfaces across the APikL effector family

The sequence diversity and crystal structures of the APikL family provide a rare opportunity to interpret information from protein structures in an evolutionary context. To determine the spatial distribution of polymorphic residues, we mapped the conservation score for each amino acid (S4 Fig; inferred by ConSurf [63]) and residues contributing to allelic diversification (Fig 9B and 9C), onto the crystal structures of AVR-Pik and APikL2, and homology models of APikL4 and APikL5. We noted that highly divergent residues (S4 Fig) primarily locate to the N-terminal region and disordered loop region of the effectors, whereas highly conserved residues tend to locate in the core structural β-sheets that contribute to the MAX-fold (S4 Fig), consistent with their role in maintaining structural integrity. Further mapping of non-synonymous mutations that contribute to allelic diversification in terminal branches of the APikL family revealed two major regions that may modulate interactions with host proteins (Figs 9C and S5).

These regions are spatially consistent with two of the three HMA binding interfaces identified in AVR-Pik/HMA complexes [7,10,43], and APikL2/HMA complexes detailed in this study. They locate to the N-terminal loop that forms interface 2 and to a surface-exposed patch formed by β-sheets 2 and 3 that corresponds to interface 1. In summary, these analyses indicate that two structurally distinct regions that map to HMA binding interfaces may contribute to adaptive evolution of the APikL family.

## Discussion

Accelerated gene evolution is a hallmark of pathogen adaptation and is often associated with host-range expansions and host-jumps [1,14–21]. However, despite extensive knowledge about the evolutionary dynamics of pathogen effectors, the molecular and biochemical basis of

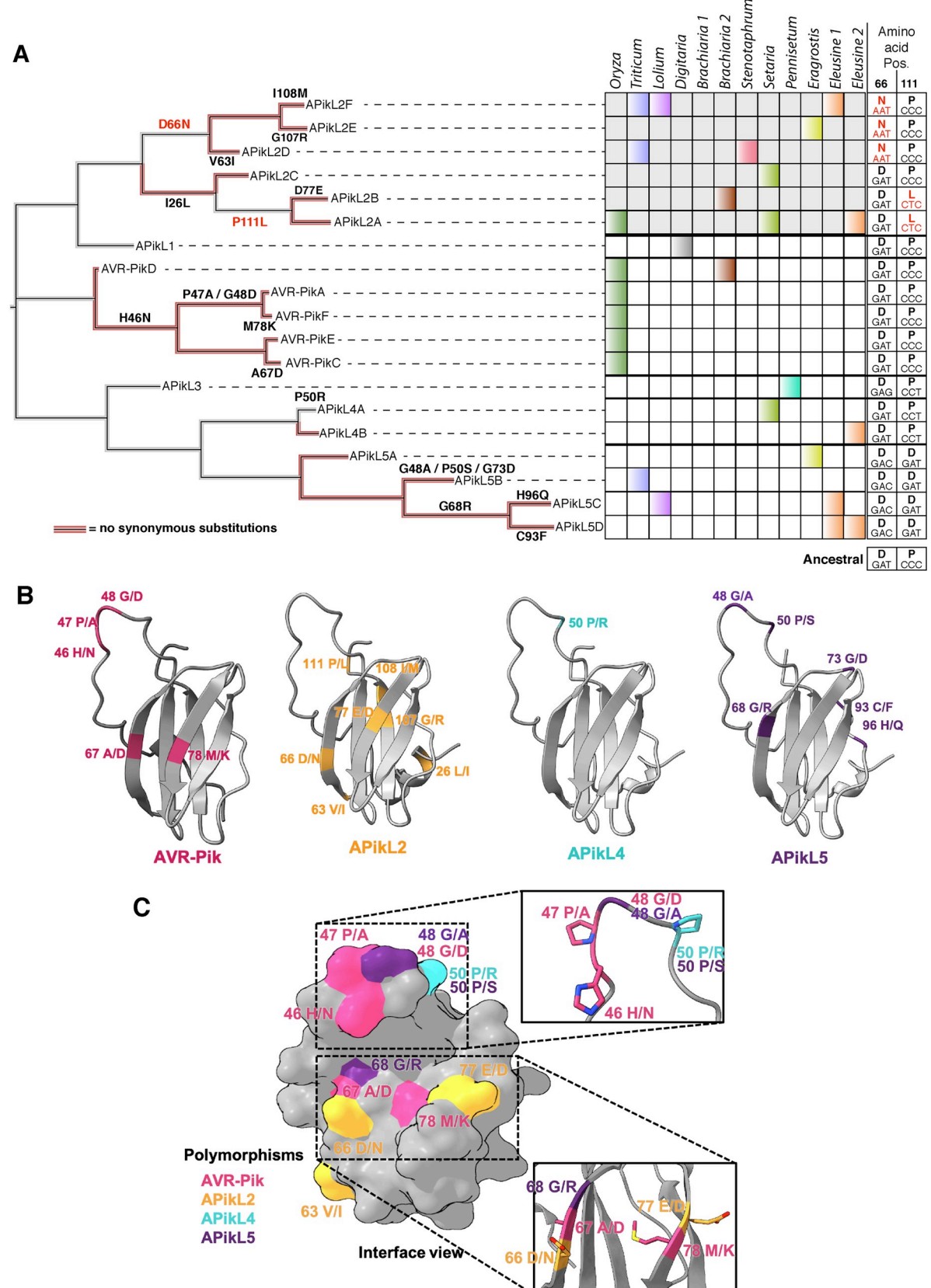

**Fig 9. Adaptive mutations shape the binding interfaces of APikL family effectors. A)** Asn-66 is a recently derived polymorphism in APikL2 effectors. Left: Reconstruction of the evolutionary history of the APikL family. Multiple non-synonymous mutations contribute to allelic diversification, largely in absence of synonymous mutations (red branches). Right: APikL effector presence/absence across multiple host-specific lineages. Colors correspond to host lineages as in Fig 1. Amino acids and codons of positions 66 and 111 across the APikL family are shown on the right. Acquired mutations are highlighted in red. The ancestral states are shown at the bottom. **B)** Mapping of non-synonymous changes that contribute to allelic diversification to the structures of APikL family members AVR-Pik, APikL2, APikL4, and APikL5. Divergent amino acids of each effector are highlighted. **C)** Allelic diversification reveals two major hotspots across the APikL family. Surface representation of divergent sites across the APikL family combining all residues shown in B. Most polymorphic residues reside in the HMA-binding interfaces 1 (bottom) and 2 (top).

how they adapt to new host-targets is still largely unknown. Here, we investigated a family of putative effector proteins with sequence similarity to the well-characterized rice blast fungus effector AVR-Pik. We found that all six members of this effector family show presence/absence polymorphisms and allelic diversification in various lineages of the multihost blast fungus. We demonstrated that allelic variants of the exceptionally conserved effector, APikL2, differentially bind the host HMA domain protein sHMA94. By combining crystal structure information of two divergent APikL2 variants in complex with HMA-domains and ancestral reconstruction of the APikL family, we showed that three APikL2 variants, APikL2D/E/F, acquired a derived mutation—Asn-66—that maps to the HMA binding interface and expands the effectors binding spectrum to the new target, sHMA94. We further showed that polymorphic residues under positive selection in the APikL family tend to locate at two major HMA-binding interfaces, suggesting that adaptation and coevolution with plant HMA domains may drive the evolution of this effector family. Taken together, our results provide a detailed molecular evolution and structural biology framework for diversification and adaptation of effectors across host-specific lineages of the blast fungus *M. oryzae*.

Effector proteins often interact with multiple members of host-target protein families [47,64–66]. In addition, multiple pathogen effectors can target the same host protein family [35,36,67], possibly to ensure effective host infection or as an evolutionary "bet-hedging" strategy to counteract detection by the plant immune system [66]. APikL2 shares the MAX-effector fold and similar HMA-binding interfaces with the well characterized effector AVR-Pik. In addition, homology modelling of APikL family members shows that most APikL effectors probably carry adaptive mutations within these binding interfaces (Fig 9). Given this, and the similar degree of sequence conservation across the APikL family (with the exception of APikL5 which contains a 19 amino acid C-terminal extension), we hypothesize that all APikL family members bind to multiple sHMA proteins via a conserved mode of action. Such a high degree of functional redundancy, along with the HMA-binding previously demonstrated for the MAX-effectors AVR-Pik, AVR1-CO39 and AVR-Pia [35], further indicates that targeting HMA proteins is an important virulence activity during host infection by the blast fungus. Future genetic analyses (gene knock-out and complementation) would need to take into account possible functional redundancy among the effectors as well as the diversity of sHMA proteins in grass hosts. In addition, we noted variation in gene expression of APikL genes in the limited public datasets that are available. In the future, comprehensive gene expression analyses of the APikL genes across multiple lineage/host combinations may reveal another layer of diversification in these effectors.

Our findings on the APikL family are consistent with the view that these effectors collectively target sHMA proteins of their multiple grass hosts with varying degrees of redundancy and specificity. We propose that diversification of the APikL family might be adaptive and driven by co-evolution with HMA containing proteins. Although the canonical member of this family, AVR-Pik, is restricted to the *Oryza*-infecting lineage of *M. oryzae*, the other five members of the family, APikL1 to APikL5, are widely distributed across other host-specific

lineages of *M. oryzae* (Figs 1 and 2). AVR-Pik, APikL2, and possibly the other APikL effectors, are structurally similar and bind HMA domains through common interfaces (Fig 9). APikL2 and AVR-Pik have overlapping spectra of HMA interactors indicating partially redundant interactions with this host target family. Interestingly, most *M. oryzae* isolates contain two members of the APikL family, with APikL2 present in isolates from all genetic lineages whereas the other APikL effectors being specific to certain host-specialized lineages. Presence of two partially redundant effector proteins might expand the spectra of sHMA proteins that are targeted during infection. In addition, the patterns of presence/absence polymorphisms observed for the APikL family across lineages of *M. oryzae* might reflect recurrent gene losses to evade plant immunity whenever one of the effectors is detected. This work sets the stage for future functional studies on the precise role of APikL effectors in host-specificity and incipient speciation of *M. oryzae*.

The six APikL2 variants grouped into two clades APikL2A-C and APikL2D-F. Particular alleles tend to be associated with a given host-specific lineage. For example, the *Oryza-*, *Setaria-*, and *Brachiaria*-infecting lineages only encode APikL2A-C and lack the other APikL2 variants, which carry the D66N polymorphism (Fig 9). Interestingly, *Eleusine*-infecting lineages are the only ones to carry both APikL2A and APikL2F variants and thus both, Asp-66 and Asn-66 (Fig 10). In addition, the *Eleusine*-infecting lineages have the most diverse repertoire of APikL effectors, carrying three APikL homologs, APikL2, APikL4, and APikL5 (Figs 1, 2, and 3). This indicates that APikL2 diversification and APikL4/5 gene gain/loss may have occurred in ancestral *Eleusine*-infecting populations prior to the emergence of the distinct genetic lineages and sorting of the various APikL genes. However, we cannot exclude recurrent host jumps between *Eleusine* and other host species as previously reported [68].

This model is further complicated by the presence of lineages of *M. oryzae* that are associated with crop plants but retain the capacity to infect wild grass hosts. For example, isolates from wild populations of *E. indica* may infect the cultivated relative *E. coracana* and vice versa (Fig 1 and [23]). Specifically, isolate PH42 which was collected from *E. coracana*, groups with *E. indica* infecting isolates in our analysis (Fig 1 and S1 Table). Conversely, isolate EI9604, collected from *E. indica* groups with isolates infecting *E. coracana* suggesting cross-infectivity between wild and cultivated relatives (Fig 1 and S1 Table). Cross-infectivity of different hosts has also been shown for multiple lineages under laboratory conditions [69]. It is possible that shuttling between wild and cultivated plants, as in between *E. indica* and *E. coracana*, could have facilitated effector diversification and such cases can serve as a springboards for host range expansions and host-jumps of the blast fungus [68]. This calls for increased surveillance of wild grass hosts, especially in regions affected by emerging blast disease [70–73], in order to understand the role they play in host-driven adaptation of *M. oryzae*.

We found that a single amino acid—Asp-66 to Asn-66—in the APikL2D-F variants leads to an expansion of the HMA-binding spectrum and that the derived Asn-66 polymorphism is likely an adaptive feature of the APikL2 variants of isolates from *Triticum*, *Lolium*, *Stenotaphrum*, *Eragrostis*, and *Eleusine*. What are the selective forces that may have driven the emergence of the Asp-66 to Asn-66 polymorphism? It is important to note that although Asn-66 only emerged in some APikL2 variants within the APikL effector family (Fig 9), it predates the split between *M. oryzae* lineages infecting *Triticum*, *Lolium*, *Stenotaphrum*, *Eragrostis*, and *Eleusine* (Fig 10). Therefore, it is difficult to ascertain which of these or any other ancestral host species may have exerted the selection pressure that led to the fixation of Asn-66. In addition, sHMA94, the newly gained host target of APikL2 variants carrying Asn-66 is from *Setaria italica*, and *Setaria*-infecting isolates do not carry Asn-66 but rather the ancestral Asp-66. Nonetheless, we propose that Asn-66 enabled APikL2 to expand its binding spectrum to new sHMA94-like proteins that probably occur in one or several of the *Triticum*, *Lolium*,

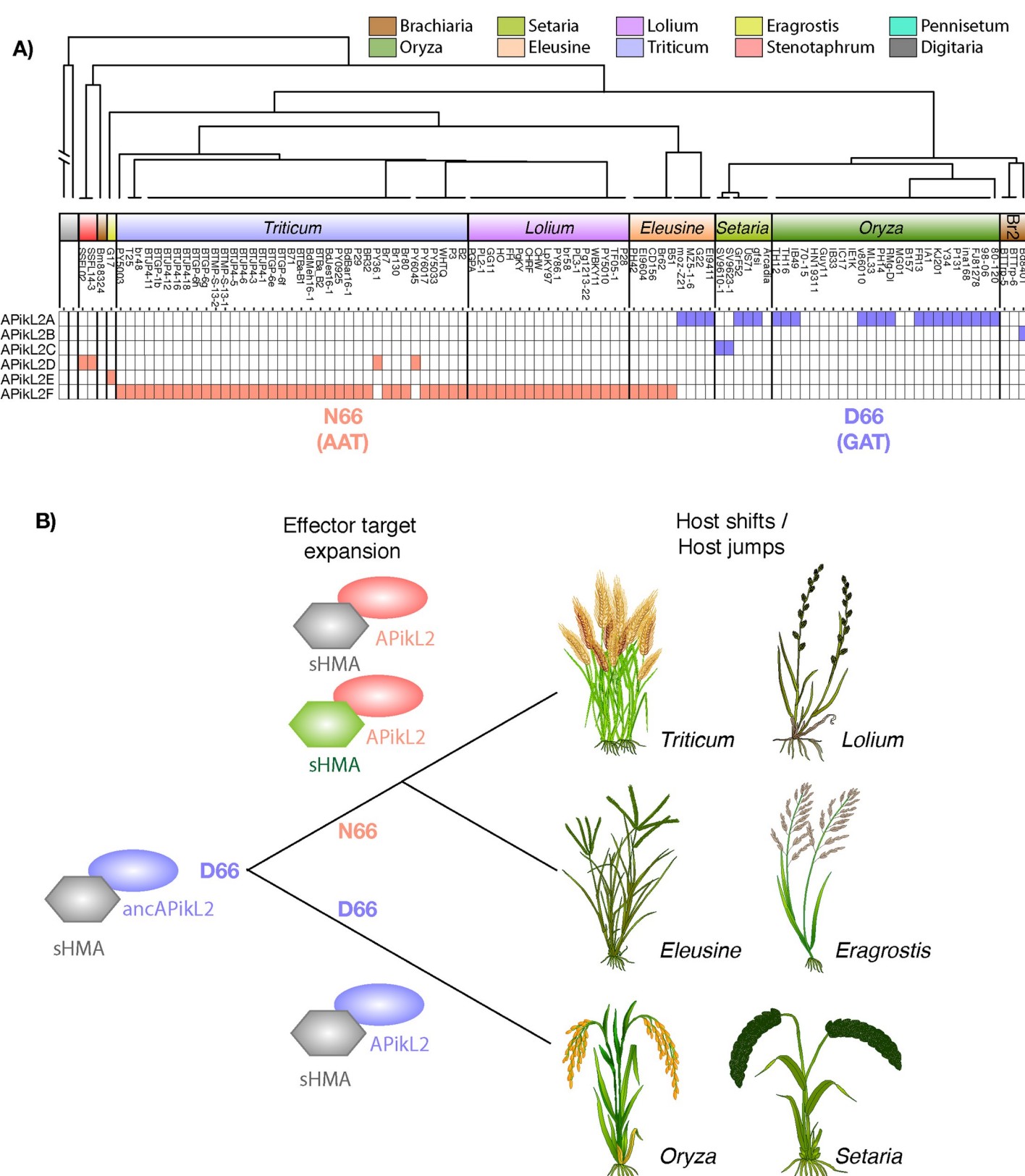

**Fig 10. The Asp-66-Asn polymorphism predates the emergence of host-adapted lineages of *M. oryzae*. A)** Distribution of the critical Asp-66 to Asn-66 mutation in *M. oryzae* populations. Schematic illustration of host-specific lineages of *M. oryzae* and annotation of the Asp-66/Asn-66 polymorphism. Blue: Asp-66; Red: Asn-66. The

host genera of the different *M. oryzae* lineages are shown. **B)** Model illustrating the emergence of the Asn-66 polymorphism prior to the differentiation of host specialized lineages. Blue: APikL2 variants carrying Asp-66; Red: APikL2 variants carrying Asn-66; Grey: sHMA target protein interacting with both APikL2 variants; Green: New sHMA target protein due to Asn-66 polymorphism.

*Stenotaphrum*, *Eragrostis*, and *Eleusine* hosts. We also conclude that it is unlikely that the Asp-66 to Asn-66 polymorphism resulted in evasion of detection by an immune receptor (*R* gene) given that it is difficult to envision how gain of binding to a new host target would evade immunity.

When we examined the structures of the APikL2/sHMA complexes we were surprised at the subtlety of the changes induced by the Asp-66 to Asn-66 polymorphism given the clear difference in sHMA94 binding we observed via yeast-2 hybrid and ITC. Comparison of the Asp-66 to Asn-66 polymorphism in the APikL2A and F alleles to the polymorphisms found in AVR-Pik alleles further highlights the discrete nature of the APikL2 polymorphism. AVR-Pik alleles that are not recognised by the Pik NLRs have reduced hydrogen bonding interactions between the effector and HMA domain as a result of their polymorphisms, which correlate with reduced affinity and loss of recognition by the receptor [7]. In contrast to this, net sum calculations of hydrogen bonds between structures of APikL2A and F bound to sHMA proteins do not explain why APikL2F has a greater affinity for sHMA94 than APikL2A, as both complexes maintain a similar number of contacts. ITC showed that APikL2A has a lower affinity for sHMA94, and the heat exchange (ΔQ) is decreased, implying significant changes in the enthalpy of the system. In correlation with this, we observe the formation of additional intramolecular hydrogen bonds around Gln-44 of sHMA94 in the presence of Asn-66 found in APikL2F, possibly reducing the mobility of Gln-44 and rigidifying this loop region of the interface. We hypothesise this rigidification of sHMA94, a result of the intramolecular hydrogen bonds in the presence of Asn-66, is intrinsic to the increased affinity of APikL2F. The hydrogen bonding network in this region may lock sHMA94 in a more favourable conformation for binding, leading to the increased affinity of the complex [74,75].

The subtlety of the molecular mechanisms underlying the APikL2F/sHMA94 interaction highlights the importance of experimental structure validation. Early attempts to predict the interactions between APikL2 and sHMA94 using homology modelling with AVR-Pik and the Pikp-1 HMA domain as templates gave little meaningful insight into the structural basis for the differential binding of APikL2A and APikL2F to sHMA94. Structural modelling of proteins to infer the impact of single residues on interactions in a complex is ultimately challenging, as side chain predictions are frequently made using backbone-dependent rotamer libraries [76,77], and do not sufficiently account for the interaction environment surrounding the residue at an interface. While protein complex modelling programs do exist, these also rely on the target maintaining significant sequence similarity with the template structure [78,79]. This presents significant hurdles for rapidly diversifying fungal effectors. While structural modelling can be used to identify putative functions and assist with categorization of sequence diverse effectors [80], characterizing the molecular details of effector-target interactions remains challenging with these approaches.

Despite major advances in understanding how pathogen effectors evade plant immunity, we still know little about how effectors acquire new host targets, especially as the pathogen shifts from one host to another [1,14–21]. Our study provides a molecular evolution and structural biology framework to study effector adaptation in a multihost pathogen. We conclude that the APikL family has diversified through a series of polymorphisms at common effector-target binding interfaces (Fig 8B and 8C). This diversification was associated with distinct patterns of genomic location with the conserved APikL2 effector residing in a highly co-linear

gene-rich region whereas the other effector genes occurring at different genomic loci in genetically distinct *M. oryzae* isolates (Fig 3). This work also illustrates the value of ancestral sequence reconstruction in yielding a more plausible model of plant-pathogen coevolution [1,43,81,82] and the unique perspectives gained from evolutionary molecular plant-microbe interactions (EvoMPMI) approaches [4,83]. By establishing the directionality of evolution and that D66 (GAT codon) is ancestral to N66 (AAT codon) (Figs 9A and 10), we can propose that the D66N polymorphism is more likely to reflect expanded host-target binding rather than evasion of host immunity. This enabled us to better understand the selection forces and historical events that have shaped the complex evolution of the APikL effector family.

## Material and methods

### Inference of a coalescence species tree by ASTRAL

To infer the species tree, we first extracted all conserved single copy orthologs from 108 genomes, representative of ten host-specific lineages, using BUSCO (version 3) with the sordariomycetes_odb9 database. Isolate PY6025 was excluded from further analysis due to low BUSCO completeness. We then used all 1920 universally conserved single copy orthologs to generate individual genealogies by RAxML (version 8.1.11) using the GTR-GAMMA model with 1000 bootstrap replications. We then used each individual best tree as input for ASTRAL (version 5.7.3). Local branch support was evaluated by local quartet tree support "q1" using the -t option in ASTRAL that reports the proportion of local quartets in the set of individual genealogies that agree with the topology of the species tree.

### Identification of APikL family effectors and potential HMA-target domains

We used an iterative TBLASTN approach to identify APikL family members using 107 genomes of *M. oryzae*. In the first round, we used AVR-PikD as a query and selected all hits with 100% query coverage. This led to the identification of APikL1-4. We then repeated the search using all APikL family members as a query. Using APikL4 as query led to the identification of the more divergent effector APikL5. This approach led to the identification of 179 candidate effectors that included 10 pseudogenes. APikL family members were classified using an amino acid percent identity cut-off of >90% for variants of the same effector.

HMA target domains were identified by TBLASTN using the integrated HMA-domain of the rice sensor NLR Pikp-1 as a query. Details about the target domains, sequences and plasmids used are available in [84]. All plasmids are available through Addgene (plasmid accession numbers 168306–168456).

### Generation of the APikL family tree

To infer the APikL family tree we removed all pseudogene sequences from the analysis. We then created a multiple sequence alignment of 169 amino acid sequences using MUSCLE [85]. We then generated a maximum-likelihood tree with 1000 bootstrap replications in MEGA7 [86].

### Protein-protein interaction

Yeast 2-hybrid interaction screen. We used the Matchmaker Gold Yeast two-hybrid system (Takara Bio USA) to detect protein-protein interactions between APikL effectors and putative HMA-target domains following the manufacturer's instructions. All effector sequences were codon optimized and synthesized for expression in *E. coli* and cloned into pGADT7 plasmids.

All HMA-domains were codon optimized and synthesized for *E. coli* expression and cloned into pGBKT7 plasmids. Each individual effector-HMA combination was then co-transformed into competent Saccharomyces cerevisiae Y2H Gold cells (Takara Bio USA) and selected on SD$^{-Leu-Trp}$ (synthetic dropout) medium. Single colonies were selected and grown overnight at 28˚C and 200 rpm to an $OD_{600}$ of 1–2 in liquid SD$^{-Leu-Trp}$ medium. Cultures were then used to prepare serial dilutions of OD600 1, $1^{-1}$, $1^{-2}$, $1^{-3}$. Of each dilution 4 μl were replica plated on SD$^{-Leu-Trp}$ and SD$^{-Leu-Trp-Ade-His}$ containing the chromogenic substrate X-α-Gal. Pictures were taken after 72–96 h incubation at 28˚C. Yeast 2-hybrid screens were repeated twice with same results. Protein expression in the APikL2 mutant screen was confirmed by western blot. Therefore, cell pellets of transformants were resuspended in SDS-PAGE loading buffer and incubated at 95˚C for 10 min. After centrifugation at 13.000 rcf for 1 min supernatants were used for SDS-Page separation and western blotting. Membranes were probed with anti-GAL4-D-NA-BD (Sigma) for the HMA domains in pGBKT7 and anti-GAL4-AD (Sigma) antibodies for the APikL effectors in pGADT7.

## Expression and purification of proteins for in vitro binding studies and crystallization

To produce proteins for biophysical analyses, *Escherichia coli* (*E. coli*) SHuffle cells were transformed with goldengate compatible pOPIN expression vectors encoding sHMA94, sHMA25 and the APikL2 variants and mutants with a cleavable 6XHIS+protein GB1 domain (GB1) solubility tag under carbenicillin selection. Inoculated cell cultures were grown in autoinduction media [87] at 30˚C for 6h and 18˚C overnight and harvested by centrifugation at 5,500 *g* for 7 mins. Pelleted cells were resuspended in 50 mM HEPES pH 8.0, 500 mM NaCl, 30 mM imidazole, 50 mM glycine and 5% glycerol (buffer A1) supplemented with EDTA free protease inhibitor tablets. The cells were lysed by sonication with a VC 750 Vibra-Cell™ (Sonics) at 40% amplitude and cell debris was removed by centrifugation at 45,000 *g* for 45 mins. Initial purification of APikL2 or sHMA protein from the clarified lysate was performed by immobilised metal affinity chromatography (IMAC) through application of the clarified lysate to a $Ni^{2+}$-NTA column connected to an AKTA Xpress purification system. 6xHis+GB1-APikL2 was step-eluted with elution buffer (buffer B1; buffer A containing 500 mM imidazole) and directly injected onto a Superdex 75 26/60 gel filtration column pre-equilibrated in 10 mM HEPES pH 8, 150 mM NaCl (buffer A4). Fractions containing 6xHis+GB1-APikL2 (as assessed by SDS-PAGE) were pooled and the 6xHis-GB1 tag was cleaved by addition of 3C protease (10 μg/mg fusion protein) with overnight incubation at 4˚C. Cleaved APikL2 or sHMA protein was purified from the digest using a $Ni^{2+}$-NTA column to perform reverse-IMAC, collecting the eluate and concentrating before a second round of size-exclusion. Proteins were then concentrated via 3 kDa cut-off spin concentrations to a final concentration of 10 mg/mL as calculated by absorbance at 205 by Nanodrop (Thermofisher).

For crystallization of the sHMA/APikL2 complexes, APikL2A and APikL2F were cloned into pOPIN-GB1 vector mentioned above. sHMA proteins were cloned into goldengate compatible pOPIN vectors untagged under a kanamycin selection. *E. coli* SHuffle cells were then co-transformed with the pOPIN vectors encoding 6HIS-GB tagged APikL2 proteins and untagged sHMA proteins and then selected for on Kanamycin 30 ug / mL + Carbenicillin 100 ug / mL LB plates. Cell cultures were then inoculated and grown as mentioned above, under Kanamycin 100 ug / mL + Carbenicillin 100 ug/ mL selection. Protein complexes were purified via IMAC coupled with size-exclusion as mentioned above. Untagged sHMA proteins were observed to be co-purified with the tagged effectors as assessed by SDS-PAGE. Confirmed protein complexes were concentrated to 10 mg/mL before storage at -80˚C.

## Crystallization, data collection and structure solution

For crystallization, APikL2/sHMA complexes were concentrated in buffer A4 (10 mM HEPES pH 7.4, 150 mM NaCl). Sitting drop, vapour diffusion crystallization trials were set up in 96-well plates, using an Oryx nano robot (Douglas Instruments). Crystallisation plates were incubated at 20˚C.

APikL2F/sHMA94 crystals typically appeared after 4 days in SG1™ Screen (Molecular Dimensions), well D10 (0.2 M lithium sulfate, 0.1 M BIS-TRIS pH 6.5, 25% PEG3350). Crystals obtained from this condition were unsuitable for diffraction and were used for cross-seeding experiments. Cross-seeding into Morpheus HT-96 screen (Molecular Dimensions) resulted in formation of crystals suitable for diffraction in condition E3 (0.12 M ethylene glycols 0.1 M buffer system 1 pH 6.5, 30% v/v precipitant mix 3). APikL2A/sHMA25 crystals suitable for diffraction were harvested after 11 days from Morpheus (Molecular Dimensions) condition H10 (0.1 M amino acids, 0.1 M buffer system 3 pH 8.5, 50% precipitant mix 4). Crystals were snap frozen in liquid nitrogen prior to shipping to the Diamond Light Source.

X-ray data sets were collected at the Diamond Light Source. The APikL2F/sHMA94 dataset was collected on the i03 beamline under proposal mx18565 and the APikL2A/sHMA25 dataset was collected on the i04 beamline under proposal mx-13467. The data were processed using the xia2 pipeline and AIMLESS, as implemented in CCP4i2 [88]. The APikL2A/sHMA25 structure was solved by molecular replacement using PHASER [89] and the PikpHMA/AVR-PikD structure. The APikL2A/sHMA25 structure was then used to solve the APikL2F/sHMA94 structure via PHASER. The final structures were obtained through iterative cycles of manual rebuilding and refinement using COOT [90] and REFMAC5 [91], and ISODLE [92], as implemented in CCP4 and ChimeraX [93] respectively. Structures were validated using the tools provided in COOT and MOLPROBITY [94]. Protein interface analyses were performed using QtPISA [62]. For each complex, one APikL2 effector/sHMA protein assembly was used as a representative example. Models are visualised using PyMol 2.0 and ChimeraX. X-ray diffraction data can be found in the Protein Data Bank (https://www.ebi.ac.uk/pdbe/) with accession numbers 7NLJ (APikL2A/sHMA25) and 7NMM (APikL2F/sHMA94).

## In vitro binding analysis with isothermal titration calorimetry

All calorimetry experiments were recorded using a MicroCal PEAQ-ITC (Malvern, UK). To test the interaction of APikL2 proteins with sHMA94, experiments were carried out in triplicate at 25˚C using buffer A4. The calorimetric cell was filled with 60 µM APikL2 and titrated with 350 uM sHMA94 protein. For each ITC run, a single injection of 0.5 µL of ligand was followed by 19 injections of 2 µL each. Injections were made at 120-second intervals with a stirring speed of 750 rpm. Difficulty with measuring concentration of the protein due to extreme molar extinction coefficients ($\varepsilon$ = 0.19 and 2.6 $M^{-1}cm^{-1}$ at $A_{280}$ for sHMA94 and APikL2 proteins, respectively) created difficulty in accurately titrating at an even stoichiometric ratio and therefore data was fit to a 1:1 binding model, with number of sites equal to one, as informed by the crystal structures of the complexes. Data were processed with AFFINImeter ITC analysis software [95]. Experimental replicates are shown in S6 Fig.

## Homology modelling and conservation predictions

Homology models of APikL4, and APikL5 were generated using Phyre2 [96] One-to-One threading with AVR-PikD (PDB ID: 6G10 chain C) as a template. Conservation of residues in AVR-Pik family members was calculated using ConSurf [97] and visualized using PyMol.

## Selection test and reconstruction of the evolutionary history of APikL family members

Genomic sequences of all APikL members were extracted from 107 *M. oryzae* genomes and multiple sequence alignments produced using MUSCLE (Edgar, 2004). Pseudogene sequences and positions with missing data (C-Terminal extension of APikL5) were removed. We then generated a codon-based, maximum likelihood phylogenetic tree with 1000 bootstrap iterations of the family using MEGA 7 (Kumar et al., 2016). We then calculated the rates of *dN/dS* ($\omega$) in the phylogenetic tree with two branch models implemented in codeml, the free-ratio model (M = 1; NSsites = 0) and the multi-ratio model (M = 2; NSsites = 0) allowing for separate ratios for each branch. The one-ratio model that allows only one *dN/dS* rate across all branches (M = 0; NSsites = 0) was rejected after performing likelihood ratio tests using the log likelihood values lnL0 = -1849.510477 and lnL1 = -1829.104349 (free ratio model) and LR = 2 x (lnL1 –lnL0) = 40.812256 (1% significance level; degrees of freedom = 18). Additionally, we calculated the pairwise synonymous (*dS*) and nonsynonymous (*dN*) substitution rates using the codeml function YN00 of PAML v4.9 (54). To detect specific sites under positive selection, we used the sire model implemented in PAML v4.9 by comparing the null models M0 (one-ratio), M1 (nearly neutral), M7 (beta), and three alternative models, M3 (selection), M2 (discrete), M8 (beta & $\omega$) and performing likelihood ratios tests as described above. Model M3 and M8 passed the chi-square test at a significance level of 1% (S2 Table). Ancestral state reconstruction was performed by a codon based maximum likelihood analysis with codeml (included in the PAML v4.9 package) according to [53,54].

## Supporting information

**S1 Table. Isolates and genomic sequences used in this study.**
(XLSX)

**S2 Table. Functional annotation of B71 gene models at APikL loci.**
(XLSX)

**S3 Table. Summary of the yeast two-hybrid interaction screen.**
(XLSX)

**S4 Table. Summary of X-ray data collection and refinement.**
(DOCX)

**S5 Table. Summary of the QtPISA interface analysis.**
(XLSX)

**S1 File. Fasta file of all APikL amino acid sequences identified in this study.**
(FAA)

**S2 File Fasta file of all intact APikL nucleotide sequences.**
(FNA)

**S1 Fig. The APikL2 locus is conserved on chromosome 3 across host-specific lineages of *M. oryzae*.** Whole genome alignments between the 70–15 reference genome assembly (Ensembl-Fungi assembly MG08) and seven isolates representative of five genetic lineages. Homologous chromosomes are color coded based on their best match in 70–15. Colored links between chromosomes show alignments >10 kb. Grey links show alignments in reverse direction (inversions). The location of each APikL gene is indicated.
(TIF)

**S2 Fig. Detailed assignment of electrostatic interactions in the APikL2A/sHMA25 and APikL2F/sHMA94 complexes.** Cartoon representation of the two effector/sHMA complexes with ball and stick rendering of interacting residues as determined by PISA (EMBL-EBI). Hydrogen bonds are represented as a yellow dashed line with cut-off distance of 4 Å. A) Interface 1 B) Interface 2 C) Interface 3.
(TIF)

**S3 Fig. The atomic interactions between Asp/Asn-66 and Gln-43/44 which facilitate the reorganisation of hydrogen bonding at the HMA binding interface.** A, B) The hydrogen bonding networks of the APikL2A/sHMA25 and APikL2F/sHMA94 complexes around Gln-43/44, respectively. A) The OD1 atom of Asp-66 of APikL2 interacts with the NE2 atom of Gln-43 of sHMA25, preventing it from interacting with the OG1 atom and backbone carbonyl of Thr-37. B) The ND2 atom of Asn-66 interacts with the OE1 atom of Gln-44, allowing the NE2 atom of Gln-44 to interact with the OG1 atom and backbone carbonyl of Thr-38. C) Superimposition of the two APikL2/sHMA complexes. The blue and red arrows adjacent to Gln-43/44 indicate the movement of the NE2 and OE1 atoms between the two structures, exemplifying the rotation of the Gln-44 residue in the APikL2F/sHMA94 complex. Similarly, the red arrow adjacent to Thr-37/38 shows the movement of the OG1 atom in the APikL2F/sHMA94 complex, due to the interaction with the NE2 atom of Gln-44.
(TIF)

**S4 Fig. Residues under diversifying selection locate to the loop regions and the N-terminal helix.** Conservation scores (determined by ConSurf) of amino acid residues in the APikL-family mapped to the crystal structure of APikL2. Colors indicate the conservation score and confidence interval.
(TIF)

**S5 Fig. The HMA-binding interface of APikL-family members is highly polymorphic.** Schematic representation of the APikL family including all polymorphic sites. APikL effectors described in this study are highlighted in bold. The crucial polymorphism in APikL2 is highlighted in red. APikL family members in grey were not polymorphic in the population studied here. Asterisks indicate polymorphisms in the signal peptide.
(TIF)

**S6 Fig. Experimental replicates of in vitro binding affinity measurements by ITC.** The top panels show heat differences upon injection of the effector into the cell containing sHMA94. The second panel shows integrated heats of injection (•) and the best fit (red line) to a single site binding model calculated using AFFINImeter ITC analysis software. The final panel shows the difference between the fit to a single site binding model and the experimental data; the closer to zero indicates stronger agreement between the data and the fit.
(TIF)

## Acknowledgments

We thank Clare Stevenson and Dave Lawson from the John Innes Centre protein X-ray crystallography platform for technical assistance during protein crystal preparation and data collection. We further want to thank all members of the BLASTOFF team at the Sainsbury Laboratory and John Innes Centre as well as Ki-Tae Kim and Yong-Hwan Lee for valuable discussions.

## Author Contributions

**Conceptualization:** Mark J. Banfield, Sophien Kamoun, Thorsten Langner.

**Data curation:** Adam R. Bentham, Yohann Petit-Houdenot, Joe Win, Mark J. Banfield, Sophien Kamoun, Thorsten Langner.

**Formal analysis:** Adam R. Bentham, Yohann Petit-Houdenot, Joe Win, Izumi Chuma, Sophien Kamoun, Thorsten Langner.

**Funding acquisition:** Mark J. Banfield, Sophien Kamoun.

**Investigation:** Adam R. Bentham, Yohann Petit-Houdenot, Mark J. Banfield, Thorsten Langner.

**Methodology:** Joe Win, Mark J. Banfield, Thorsten Langner.

**Project administration:** Sophien Kamoun.

**Resources:** Izumi Chuma, Ryohei Terauchi, Thorsten Langner.

**Supervision:** Mark J. Banfield, Sophien Kamoun, Thorsten Langner.

**Validation:** Adam R. Bentham, Thorsten Langner.

**Visualization:** Adam R. Bentham, Sophien Kamoun, Thorsten Langner.

**Writing – original draft:** Adam R. Bentham, Sophien Kamoun, Thorsten Langner.

**Writing – review & editing:** Adam R. Bentham, Yohann Petit-Houdenot, Joe Win, Izumi Chuma, Ryohei Terauchi, Mark J. Banfield, Sophien Kamoun, Thorsten Langner.

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
