## [Decision Letter · Decision Letter 0]

27 May 2021

Dear Prof. Kamoun,

Thank you very much for submitting your manuscript "A single amino acid polymorphism in a conserved effector of the multihost blast fungus pathogen expands host-target binding spectrum" for consideration at PLOS Pathogens. As with all papers reviewed by the journal, your manuscript was reviewed by members of the editorial board and by several independent reviewers. In light of the reviews (below this email), we would like to invite the resubmission of a revised version that takes into account the reviewers' comments, particularly those from Reviewer #1.

We cannot make any decision about publication until we have seen the revised manuscript and your response to the reviewers' comments. Your revised manuscript is also likely to be sent to reviewers for further evaluation.

Sincerely,

Jin-Rong Xu, PhD

Associate Editor

PLOS Pathogens

Brett Tyler

Section Editor

PLOS Pathogens

Kasturi Haldar

Editor-in-Chief

PLOS Pathogens

orcid.org/0000-0001-5065-158X

Michael Malim

Editor-in-Chief

PLOS Pathogens

orcid.org/0000-0002-7699-2064

Reviewer's Responses to Questions

**Part I - Summary**

Reviewer #1: Arms race co-evolution is predicted to drive genetic diversification in plant-pathogen interactions. A multitude of studies have demonstrated increased variation in genes encoding virulence factors of plant pathogens. Nevertheless, very few studies have been able to experimentally demonstrate the role of genetic variation in altering the outcome of protein-protein interactions in plant-pathogen systems. The study by Bentham and colleagues hereby provides a convincing and important example of protein-protein interaction and co-evolution in a plant-pathogen system.

The authors have investigated allelic variation in paralogs of the effector gene Avr-PIK in 107 genomes of the plant pathogenic fungus Magnaporthe oryzae. The study integrates sequence analyses, molecular assays and structural predictions. Previous studies have described variation in Avr-Pik and associated allelic variants to distinct host specificities. Also the role of Avr-Pik in binding to heavy metal-associated (HMA) domains of immune receptors has been described in earlier studies. The novelty of the present study is the precise mapping of a single mutation that confers a change in binding affinity to HMA domains. Moreover, the study presents evolutionary predictions to identify sites under selection and to reconstruct the ancestral state of the particular site, as well as the entire protein. Together these analyses provide a convincing and unique example of molecular plant-pathogen co-evolution.

Reviewer #2: This study reports the remarkable specificity of fungal effector proteins in host protein binding and exploration of multiple variants of these effectors across a collection of M. oryzae strains. The evolutionary analyses indicate that positive selection has shaped the evolution of these proteins and the specific residues and their function are interpreted in the context of the crystal structure of the protein.

The work is really impressive and posits a hypothesis that one codon change has modified binding affinity to a host factor which is correlated to host shift in isolates. I think this is intriguing, worthy of publication, and establishes a new understanding of the ongoing host adaptation evolutionary processes in Magnaporthe.

**Part II – Major Issues: Key Experiments Required for Acceptance**

Reviewer #1: While I enjoyed reading the manuscript, some issues should still be addressed:

Previous studies have demonstrated the functional relevance of Avr-Pik and other HMA-binding effectors in M. oryzae. I understand that the diversity of these effectors reflects a large redundancy in the effector repertoire of this pathogen. However, focusing on APikL2, what is the relevance of this effector in the outcome of pathogen infection? In the Discussion, the authors speculate that APikL effectors could play a role in speciation which would imply a significant role as virulence determinants. A deletion mutant would allow the authors to assess the relevance of the APikL2 gene in virulence and host specificity.

The ancestral sequence reconstruction analysis is important to assess which mutation represent the recently derived one. However, the authors provide no details to the methods that they have applied for the ancestral sequence reconstructions. It is unclear if this was done using parsimony or maximum likelihood analyses? If it was conducted at the codon or amino acid level? Detailed information to the applied methods must be provided.

Given the dataset, I am concerned about the application of the “site model” to identify sites under positive selection. If I understand correctly, the authors have included both paralogs and orthologs in their analyses. The site model assumes that all sequences evolve under the same regime, that is, selective pressure varies among sites and not among branches. In this case, I would recommend doing a branch-site model instead, using the candidate branch to define a foreground clade to test for sites under positive selection in this clade only.

There are several transcriptome studies of M. oryzae. Do the authors know if the APikL genes are expressed (also in the case of more than one gene)?

Reviewer #2: The genomic locations of the ApiKL family members don't seem to be discussed. Are these dispersed copies or do they occupy syntenic or idiomorphic locations among the genomes? I was not clear from the descriptions what was the scenario.

**Part III – Minor Issues: Editorial and Data Presentation Modifications**

Reviewer #1: Page 3, upper paragraph: for readers outside the field of plant pathology, it would be good to define “biotrophic”.

Page 4: please write out terms for abbreviations “encoded by R or Pi genes”

Page 7: How were the genes recognized as pseudogenes? Do these pseudogenes share some of the same mutations?

Fig. 3A: It is interesting that ApikL5 also exhibits signatures of selection. Which sites are identified to be under selection in the ApikL5 branch? It would be interesting to provide this information in relation to the patterns in APikL2.

Fig. 3B: The yellow dots are hardly visible. I recommend using another colour.

Fig. 4A: It would be helpful to provide the amino acid positions on the top of the left schematic diagram.

Reviewer #2: The classification of AVR-PikL members into familes is a little unclear to me from the methods and the results section. The TBLASTN approach took a query (AVR-Pik) and found homologs. But how do you know for strain B51 and moz-Z21 that the hits are AVR-PikL4 or AVR-PikL5? was a % identity cutoff used? was a tree used to classify the genes into their sub-families? Figure 1 shows the representation of the copies but I am not able to see how these were operationally assigned.

It seems like this is clearly shown in gene tree in Figure 2 - the tree confirms the assignments but I am not seeing if this gene tree information was used to make sub-family classifications?

Is the Gene tree rooted? if you root it with a copy from a sister species -- I was going to look for this but I had trouble initially finding the ApiK gene named in databases, it would be useful if the Magnaporthe oryzae locus name was used somewhere in the paper so that links to databases like FungiDB could be directly accessed from this paper. Perhaps specifying the protein accession number and locus ID of the query ApiK protein in the Methods could be provided.

Figure 3- It is unclear why the dN axis needs to be as tall as dS - it would be easier to see the detail if truncated at 0.8? Not critical just a style suggestion.

Please provide more detail in the methods for reproducibility of the computational analyses. Versions of programs (BUSCO, ASTRAL,RAxML) should be reported.

"Inference of a coalescence species tree by ASTRAL" -> please list which BUSCO marker set you used - sordariomycetes_odb10? ascomycota_odb10 ?

Did the authors extract protein sequences of BUSCO markers - generally it doesn't return codiing sequences from BUSCO so I was not sure how they got nucleotide trees from the BUSCO analyses? Maybe the authors can elaborate.

Be consistent in tool name - TBLASTN is the tool name but I see it as tblastn and tBlastN in places.

The TBLASTN approach to generate gene models is appropriate but how was splicing confirmed? Usually tools like exonerate are better at getting splice boundaries and producing a coding sequence and proteins for downstream analysis. Was the protein generated from TBLASTN for the gene trees cleaned up in any way or just taking the translated sequence reported from TBLASTN?

page 26 "effectors that included 10 pseudogenes." and page 27:

"To infer the APikL family tree we removed all pseudogene sequences from the analysis" - please detail how you identified pseudogenes?

Langner et al Zenodo 2021 - I don't see a DOI so was unable to verify what details of the data are deposited there (but I applaud the authors for making all supporting scripts and data available in a citeable archive)

More detail could be added to the description of the APikL gene tree - "general topology of host species was inferred using NCBI" - do you mean this was a constraint? How was this topology integrated with the MEGA generated ML tree?

For the selection analyses (page 30) - I think the authors used the SITE model not the SIRE model - typo?

Ensembl not Ensemble (page 12, 13)

PLOS authors have the option to publish the peer review history of their article (what does this mean?). If published, this will include your full peer review and any attached files.

Reviewer #1: No

Reviewer #2: No
---

## [Decision Letter · Decision Letter 1]

14 Sep 2021

Dear Prof. Kamoun,

We are pleased to inform you that your manuscript 'A single amino acid polymorphism in a conserved effector of the multihost blast fungus pathogen expands host-target binding spectrum' has been provisionally accepted for publication in PLOS Pathogens.

Best regards,

Jin-Rong Xu, PhD

Associate Editor

PLOS Pathogens

Brett Tyler

Section Editor

PLOS Pathogens

Kasturi Haldar

Editor-in-Chief

PLOS Pathogens

orcid.org/0000-0001-5065-158X

Michael Malim

Editor-in-Chief

PLOS Pathogens

orcid.org/0000-0002-7699-2064

Reviewer Comments (if any, and for reference):

Reviewer's Responses to Questions

**Part I - Summary**

Reviewer #1: The authors have responded to my previous comments as well as comments provided by a second reviewer, and they have prepared a much-improved manuscript.

The new detailed analyses of synteny around the APikL2 locus as well as around the other APik genes are interesting. These new analyses complement Figure 1 also showing higher extent of conservation of APIKL2 in terms of presence/absence variation.

Page 11 and Figure 3 summarize the new analyses of synteny. The text appears not so well-worked through as the rest of the manuscript. It reads a bit long and the final conclusion “the genomic location of APikL2 is more conserved than the regions encoding the other APikL genes” is phrased two times, with almost the same wording (end of first paragraph page 11 and page 12). The authors should shorten the text of these new analyses.

In the figure legend of Fig 3A and B it is not mentioned how the phylogenetic relationship between the M. oryzae chromosomes/scaffolds was determined.

The pipelines used for the alignment and synteny analyses are not mentioned in the Materials and Methods section

Reviewer #2: The revision addressed the raised concerns by both reviewers.

The improved figures and clarity to the points raised helped improve the paper and presentation.

**Part II – Major Issues: Key Experiments Required for Acceptance**

Reviewer #1: (No Response)

Reviewer #2: (No Response)

**Part III – Minor Issues: Editorial and Data Presentation Modifications**

Reviewer #1: (No Response)

Reviewer #2: (No Response)

PLOS authors have the option to publish the peer review history of their article (what does this mean?). If published, this will include your full peer review and any attached files.

Reviewer #1: No

Reviewer #2: **Yes: **Jason Stajich

---

## [Editor Report · Acceptance letter]

5 Nov 2021

Dear Prof. Kamoun,

We are delighted to inform you that your manuscript, "A single amino acid polymorphism in a conserved effector of the multihost blast fungus pathogen expands host-target binding spectrum," has been formally accepted for publication in PLOS Pathogens.

Best regards,

Kasturi Haldar

Editor-in-Chief

PLOS Pathogens

orcid.org/0000-0001-5065-158X

Michael Malim

Editor-in-Chief

PLOS Pathogens

orcid.org/0000-0002-7699-2064